# Targeting Tyro3 ameliorates a model of PGRN-mutant FTLD-TDP via tau-mediated synaptic pathology

Kyota Fujita[1], Xigui Chen[1], Hidenori Homma[1], Kazuhiko Tagawa[1], Mutsuki Amano [2], Ayumu Saito[3], Seiya Imoto [4], Hiroyasu Akatsu[5], Yoshio Hashizume[6], Kozo Kaibuchi[2], Satoru Miyano[3] & Hitoshi Okazawa[1]

Mutations in the progranulin (*PGRN*) gene cause a tau pathology-negative and TDP43 pathology-positive form of frontotemporal lobar degeneration (FTLD-TDP). We generated a knock-in mouse harboring the R504X mutation (PGRN-KI). Phosphoproteomic analysis of this model revealed activation of signaling pathways connecting PKC and MAPK to tau prior to TDP43 aggregation and cognitive impairments, and identified PKCα as the kinase responsible for the early-stage tau phosphorylation at Ser203. Disinhibition of Gas6 binding to Tyro3 due to PGRN reduction results in activation of PKCα via PLCγ, inducing tau phosphorylation at Ser203, mislocalization of tau to dendritic spines, and spine loss. Administration of a PKC inhibitor, B-Raf inhibitor, or knockdown of molecules in the Gas6-Tyro3-tau axis rescues spine loss and cognitive impairment of PGRN-KI mice. Collectively, these results suggest that targeting of early-stage and aggregation-independent tau signaling represents a promising therapeutic strategy for this disease.

[1] Department of Neuropathology, Medical Research Institute and Center for Brain Integration Research, Tokyo Medical and Dental University, 1-5-45 Yushima, Bunkyo-ku, Tokyo 113-8510, Japan. [2] Department of Cell Pharmacology, Graduate School of Medicine, Nagoya University, 65, Tsurumai, Showa, Nagoya, Aichi 466-8550, Japan. [3] Human Genome Center, Institute of Medical Science, The University of Tokyo, 4-6-1, Shirokanedai, Minato-ku, Tokyo 108-8639, Japan. [4] Health Intelligence Center, Institute of Medical Science, The University of Tokyo, 4-6-1, Shirokanedai, Minato-ku, Tokyo 108-8639, Japan. [5] Department of Medicine for Aging in Place and Community-Based Medical Education, Nagoya City University Graduate School of Medical Sciences, Nagoya, Aichi 467-8601, Japan. [6] Department of Neuropathology, Institute for Medical Science of Aging, Aichi Medical University, 1-1 Yazakokarimata, Nagakute, Aichi 480-1195, Japan. Kyota Fujita and Xigui Chen contributed equally to this work. Correspondence and requests for materials should be addressed to H.O. (email: okazawa-tky@umin.ac.jp)

Progranulin (PGRN) was initially described as a precursor protein that is cleaved to produce multiple forms of granulin (also known as epithelin, GRN), which either promote or inhibit cell growth[1–3]. PGRN itself has mitotic effects on various types of cells, including neural stem/progenitor cells (NSPCs). In mature neurons, PGRN promotes neurite extension[4], and deficiency of this factor decreases gross neural connectivity in vivo[5]. In mice, PGRN deficiency also triggers neuroinflammation and increases vulnerability to 1-methyl-4-phenyl-1,2,3,6-tetrahydropyridine (MPTP)-induced Parkinsonism[6]. A recent study showed that PGRN suppresses C1qa-dependent circuit-specific synaptic pruning by microglia; consequently, PGRN deficiency in homozygous knockout mice induces activation of microglia from 16 weeks of age, leading to the loss of VGAT-positive inhibitory synapses and elevated abundance of VGlut2-positive excitatory synapses from 32 weeks of age[7].

In humans, mutations in *PGRN* are associated with familial frontotemporal lobar degeneration (FTLD) and amyotrophic lateral sclerosis (ALS) with pathology of transactive response DNA-binding protein 43 kD (TDP43)[8–12]. The sites of clinically detected mutations are distributed throughout the molecule, implying that haploinsufficiency of PGRN due to nonsense-mediated RNA decay, rather than loss of function of a specific GRN, might be responsible for FTLD[13]. A null mutation of *PGRN* was reported in a sporadic case of FTLD[14], and DNA methylation of the *PGRN* promoter is altered in some sporadic FTLD cases, resulting in reduced expression[15].

TDP43 is a major component of neuronal aggregates in tau-negative FTLD[16,17]. Patients with mutations in PGRN develop FTLD with TDP43 aggregation (FTLD-TDP), which is pathologically similar to the result of TDP43 mutation[18]. Therefore, TDP43 is assumed to be the downstream effector of PGRN in this type of FTLD. However, it remains unclear whether aggregation of TDP43 is indispensable for the initiation of pathology. Because TDP43 is an intrinsically denatured (or disordered) protein that forms nuclear or cytoplasmic bodies through self-aggregation, mutations affect its dynamism and physiological functions rather than generating solid aggregates of TDP43 corresponding to the long fibrils observed at the initial stage of FTLD[19–21]. Moreover, it is not known that molecules initiate the pathology prior to TDP43 aggregation, and it remains unclear how functional changes in synapses occur in FTLD.

To investigate the molecular mechanisms of PGRN-linked FTLD, several groups generated *Pgrn* knockout (PGRN-KO) mice[6,22–27], which exhibit exaggerated inflammation, cellular aging, accelerated ubiquitination, elevated caspase activation, and reduced TDP43 solubility. Insufficient inhibition of microglia activation has been suggested to promote pruning of spines of inhibitory neurons in PGRN-KO mice[7].

However, as often pointed out in discussions of animal models of neurodegenerative diseases, including Alzheimer's disease (AD)[28], both copies of the *Pgrn* gene are artificially ablated in the homozygous PGRN-KO mouse model[7]. In contrast to the homozygotes, the heterozygous KO mice do not exhibit obviously abnormal phenotypes, probably due to unnatural expression and/or metabolism of PGRN that differs from the human pathology.

In this study, we generated a mutant *Pgrn* (R504X) knock-in mouse model (PGRN-KI) that successfully mimics TDP43 pathology and recapitulates the associated progressive cognitive impairment. Using this new model, we identified a new phosphorylation site of tau that is linked to initiation of synapse pathology prior to TDP43 aggregation, as well as other pathological events such as microglial activation. Moreover, we discovered that PGRN inhibits the interaction of Gas6 with the TAM family receptor tyrosine kinase Tyro3. The reduction in the PGRN level in the mutant mice activated Tyro3 signaling, leading to PKC and MAPK activation, mislocalization of Ser203-phosphorylated tau, and reduction in the number of synaptic spines.

All of these pathological events occurred before TDP43 aggregation in the brain. Collectively, our findings reveal a new tau phosphorylation–dependent mechanism, initiated before TDP43 aggregation that plays critical roles in the pathology of non-tau FTLD.

## Results

### PGRN-KI mice exhibit phenotypes resembling human FTLD.

In the C57BL/6J background, we generated mutant *Pgrn* knock-in mice harboring the R504X mutation (PGRN-KI). This point mutation corresponds to the human R493X mutation causally linked to PGRN-linked FTLD[13,14]. The mutation predominantly causes dementia rather than motor neuron disease or other symptoms[13,14].

We performed a detailed analysis of brain pathology in heterozygous PGRN-KI mice. PGRN-linked FTLD, classified as FTLD-TDP[29], is characterized by nuclear and cytoplasmic aggregation or cytoplasmic translocation of TDP43, a nuclear protein involved in RNA processing[16,17]. Anti-TDP43 and anti-phospho-TDP43 antibodies clearly detected cytoplasmic inclusion bodies, lentiform intranuclear inclusions, and cytoplasmic staining of TDP43 in mice from 24 weeks of age (Fig. 1a). The sarkosyl-insoluble fraction prepared from whole cerebral cortex of PGRN-KI mice at 24 weeks of age contained phosphorylated TDP43 (Fig. 1b). Consistent with this, cytoplasmic and nuclear aggregates were stained with anti-Ub antibody in PGRN-KI mice at 24 and 48 weeks of age (Fig. 1c). The proportions of neurons possessing TDP43-positive and Ub-positive cytoplasmic aggregates increased over the course of aging in PGRN-KI mice, especially in the cerebral cortex (Fig. 1d).

Western blot analysis using two independent antibodies against PGRN revealed that the level of full-length PGRN protein was reduced in cerebral cortex of PGRN-KI mice (Supplementary Fig. 1A). Quantitative PCR revealed a corresponding reduction in the level of *Pgrn* mRNA (Supplementary Fig. 1B) due to nonsense-mediated decay (Supplementary Fig. 1C).

Body weight was lower in PGRN-KI mice than in wild-type mice of the same genetic background (C57BL/6J) from birth until 16 weeks of age (Supplementary Fig. 1D, E), but recovered in the mutant animals by the age of 20 weeks (Supplementary Fig. 1D). Brain weight was also slightly lower in PGRN-KI mice than in controls (Supplementary Fig. 1F), but no structural abnormalities were observed (Supplementary Fig. 1G).

We subjected the mice to six behavioral tests: open field, light–dark box, elevated plus maze, rotarod, fear conditioning, and Morris water maze. In the fear-conditioning test, the PGRN-KI mice exhibited a statistically significant decrease in total freezing time from 12 weeks of age (Supplementary Fig. 1H). In the Morris water maze test, 12-week-old PGRN-KI mice exhibited a similar decrease in time spent at the target region or number of target crosses in comparison with wild-type animals (Supplementary Fig. 1I). Together, these observations indicate that onset of symptoms had occurred by 12 weeks of age. No abnormalities were detected in the rotarod or other tests until 24 weeks (data not shown). These test outcomes largely correspond to the clinical symptoms of human PGRN-linked FTLD patients harboring the R493X mutation[13,14].

### Early-stage changes in phospho-signaling in PGRN-KI mice.

To detect early pathological signatures, we performed comprehensive phosphoproteome analyses of cerebral cortex tissues and investigated signaling pathways specifically activated in the brains

of PGRN-KI mice. In these experiments, we subjected cerebral cortex tissue from three PGRN-KI and three control (C57BL/6J) mice to comprehensive phosphoproteome analysis on an AB SCIEX 5600 instrument. In brief, phosphopeptides were enriched using the Titansphere Phos-Tio Kit (GL Sciences Inc.) and

iTRAQ-labeled using the iTRAQ Reagent multiplex assay kit (AB SCIEX). By improving the sample preparation method and fractionating samples by cation exchange chromatography, we were able to detect 40,000–90,000 peptides and 13,000–25,000 phosphopeptides at > 95% confidence in PGRN-KI and control

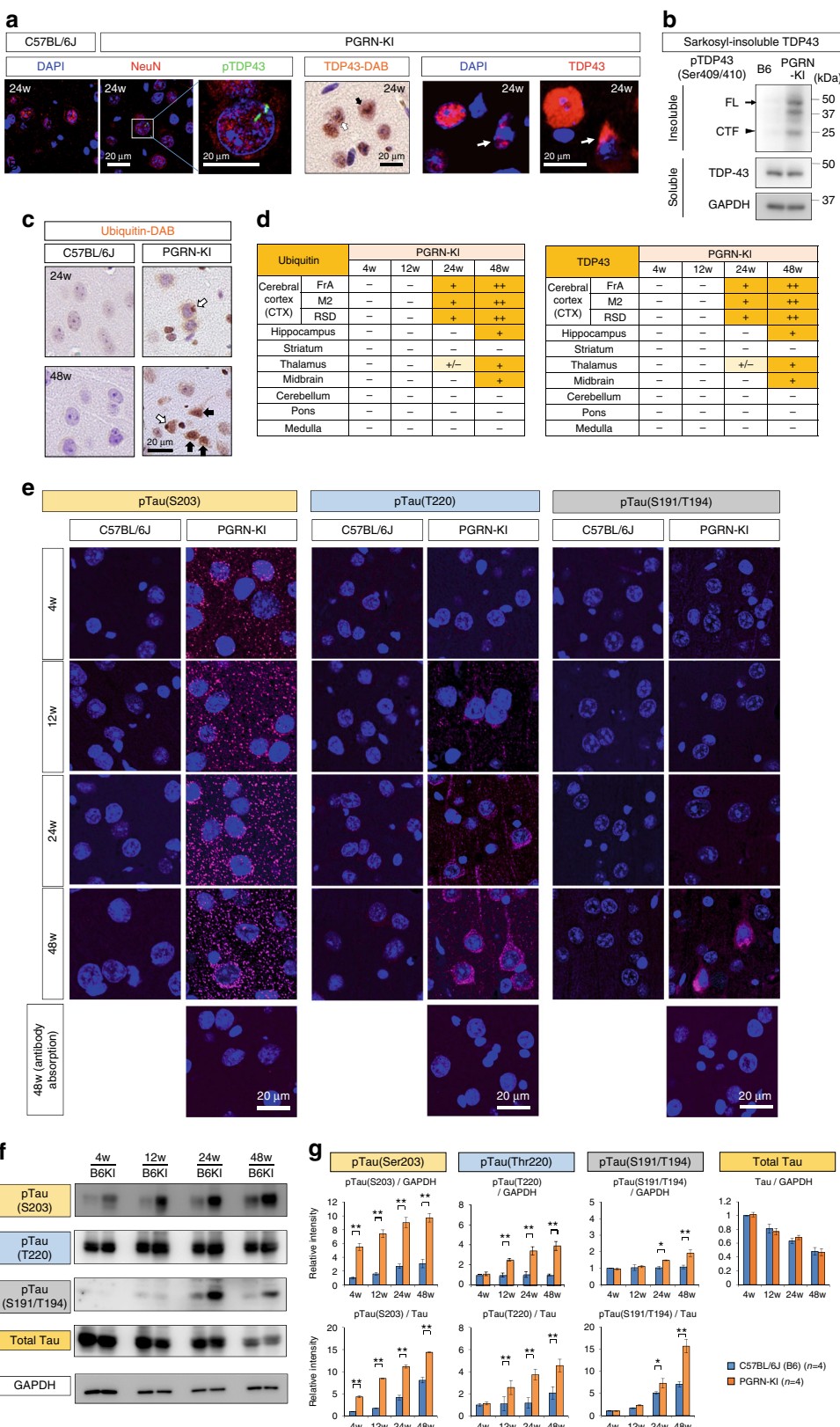

mice (Supplementary Fig. 2A, upper table). This led to identification of 1000–1900 independent proteins, 640–1100 independent phosphoproteins, and 600–1100 independent iTRAQ-labeled phosphoproteins at >95% confidence (Supplementary Fig. 2A, lower table).

The iTRAQ-derived quantities of multiple peptides overlapping at each phosphorylation site were integrated, and the resultant value was considered to represent the amount of phosphorylation at that site. We calculated the ratios between PGRN-KI and C57BL/6J mice at all detected phosphorylation sites, and evaluated the statistical significance of differences using Welch's test with the post-hoc Benjamini–Hochberg correction. Next, we constructed pathological protein–protein interaction (PPI) networks based on significantly changed phosphoproteins with $q$-values <0.05 in PGRN-KI mice at 4, 12, and 24 weeks of age. In the diagram in Supplementary Fig. 2B, nodes indicate proteins with increased or decreased phosphorylation (red or green, respectively) in PGRN-KI mice and the proteins directly connected to the altered proteins in the PPI database (blue). The pathological networks changed dynamically during the time course; in particular, the number of red nodes (elevated phosphorylation) increased with age (Supplementary Fig. 2B).

Next, we investigated signaling pathways activated in PGRN-KI mice by mapping altered phosphoproteins onto the KEGG (Kyoto Encyclopedia of Genes and Genomes) pathway database (http://www.genome.jp/kegg/pathway.html; Supplementary Fig. 2C). This analysis revealed that the MAPK, mTOR, insulin, and antigen processing/presentation pathways were altered in PGRN-KI, whereas the TNF and TNF-related signaling pathways were not (Supplementary Fig. 2C). In addition, pathways related to kinases such as CaMKII, PKA, RhoK, and Cdk5 leading to tau were unchanged at 4 and 12 weeks of age (Supplementary Fig. 2C). Phosphorylation changes in PGRN-KI mice were restricted to a few kinase pathways such as the PKC and B-Raf pathways, which act upstream of tau (Supplementary Fig. 2C) and are broadly included in the insulin pathway (Supplementary Fig. 2D). Notably, no membrane receptors upstream of the PKC or B-Raf pathways were identified in this analysis, suggesting that one or more unknown membrane tyrosine kinases might activate the core signaling pathway in PGRN-KI mice (shown as 'X' in Supplementary Fig. 2D).

**Tau phosphorylation in non-tau FTLD**. Detailed analysis of phosphoproteome data at the peptide level revealed that phosphorylation of Shc, B-Raf, PKCα, PKCγ, MEK1, tau, stathmin, GSK3β, Mef2c, and CAP/Sorbs1 at specific sites was significantly elevated in PGRN-KI mice (Supplementary Fig. 3A), and that some sites were more intensely phosphorylated at multiple time points (marked with yellow in Supplementary Fig. 3A). In tau,

phosphorylation levels of only three (of many) sites (Ser203, Thr220, Ser393) were elevated at multiple time points (Supplementary Fig. 3A); these sites are conserved across species in multiple splicing isoforms of tau (Supplementary Fig. 3B).

Among the tau phosphorylation sites, immunohistochemistry confirmed elevated phosphorylation only at Ser203 and Thr220 (Fig. 1e). Anti-phospho-Ser203 antibody revealed a remarkable increase in cytoplasmic staining of neurons in PGRN-KI mice from 4 weeks of age, whereas anti-phospho-Thr220 antibody detected the increase starting at 12 weeks (Fig. 1e). Tau phosphorylation at Ser191/Thr194 detected by the AT8 antibody, used as a control, was present in a small percentage of neurons only at 48 weeks of age (Fig. 1e).

Western blot analysis with anti-phospho-Ser203 and anti-phospho-Thr220 antibodies revealed similar patterns of increase (Fig. 1f). Phosphorylation of tau at Ser203 was elevated in PGRN-KI mice from 4 weeks of age, whereas phosphorylation at Thr220 was slightly elevated starting at 12 weeks (Fig. 1f). In addition, the western blots revealed two important findings. First, the level of phosphorylation at Ser203 and Thr220 was slightly elevated during aging in wild-type sibling mice (C57BL/6J; abbreviated as B6 in figures; Fig. 1f), but the difference between B6 and PGRN-KI mice became greater, especially at Ser203, from 4 to 48 weeks of age (Fig. 1f). Consistent with the tau finding, immunohistochemistry of B-Raf, MEK, and ERK1 revealed increases in the phosphorylated forms in PGRN-KI mice from 4 weeks of age (Supplementary Fig. 3C). Second, the total amount of tau clearly decreased during aging in both B6 and PGRN-KI mice (Fig. 1f). This finding was essential to compare our hypothesis with previous data suggesting that tau was decreased in human PGRN-linked FTLD brains[30]. According to our data, the reduction in total tau was linked to aging rather than PGRN-linked FTLD pathology (Fig. 1f, g).

These findings suggested that the PKC and MAPK pathways were activated in PGRN-KI mice, leading to tau phosphorylation, even though PGRN mutations are associated with non-tau FTLD. Hence, we decided to investigate the pathological significance of tau phosphorylation in PGRN-linked FTLD.

**PGRN binds to Gas6 and inhibits the Gas6–Tyro3 interaction**. To determine how a reduction in the level of PGRN protein leads to tau phosphorylation, it was essential to identify the receptor that transfers the effect of PGRN to the MAPK pathway. Previously, we performed a computational analysis of data from a comprehensive phosphoproteome analysis of human AD patients and mouse AD models[31]. This analysis identified MARCKS, a PKC substrate protein that anchors the actin network to the plasma membrane, as a molecule that is differentially expressed at the earliest time point when no pathological amyloid aggregation

**Fig. 1** Phosphorylation of Tau in PGRN-KI mice. **a** Immunohistochemistry with anti-phospho-TDP43 antibody revealed nuclear rod-like aggregates in NeuN-positive neurons of the frontal cortex (M2) of PGRN-KI mice at 24 weeks of age, but no such aggregates were observed in C57Bl/6 J mice (left panels). Anti-TDP43 antibody with DAB (middle panel) revealed cytoplasmic stains (white arrow) and heterogeneous nuclear staining (black arrow). Anti-TDP43 antibody revealed nuclear and cytoplasmic staining, indicated by arrows (right panels). **b** Western blots for pTDP43 (pS409/410) of the sarkosyl-insoluble fraction of whole cerebral cortex from B6 and PGRN-KI mice at 24 weeks of age. Insoluble fractions contained full-length (FL) and C-terminal fragment (CTF) of phosphorylated TDP43. **c** Ubiquitin staining revealed cytoplasmic aggregates at 24 weeks (white arrow), and the abundance of nuclear aggregates increased at 48 weeks of age (black arrow). Representative images were taken from M2. **d** Summary of TDP43 and ubiquitin staining in various regions at the indicated time points. '+', 1–4 positive cells/field; '++', 5 or more positive cells/field (field size: 430 × 550 μm). **e** Three types of anti-phospho-tau antibodies yielded different stain patterns in frontal associated cortex (FrA) of PGRN-KI mice. Anti-phospho-Ser203-tau antibody stained neuropils of PGRN-KI mice from 4 weeks of age, whereas anti-phospho-Thr220-tau antibody stained cytoplasm and neurites of PGRN-KI mice from 12 weeks of age. Anti-phospho-Ser191/Thr194-tau (AT8) antibody stained cytoplasm of cortical neurons only at 48 weeks. The similar chronological changes were observed in M2 and RSD. Antibody absorption was performed using each phosphorylated tau peptide. **f** Western blots with anti-phospho-Ser203-tau, anti-phospho-Thr220-tau, and anti-phospho-Ser191/Thr194-tau (AT8) antibodies. **g** Quantitative analyses of western blots with four mice at each time point. *$p < 0.05$; **$p < 0.01$ ($N = 4$, Student's $t$-test). Averages and s.e.m. are shown. $p$-values are shown in Supplementary Data 1

is detectable in the brain[31]. Because our phosphoproteome analysis in this study revealed an increase in MARCKS phosphorylation in PGRN-KI mice (Supplementary Fig. 3D), we speculated that a membrane receptor associated with MARCKS would be a candidate receptor for PGRN. The String PPI database (http://string91.embl.de/, http://string-db.org/) indicated that Tyro3, a TAM family receptor[32], is associated with MARCKS (Fig. 2a). TAM receptors (including Tyro3) indirectly activate PKC via

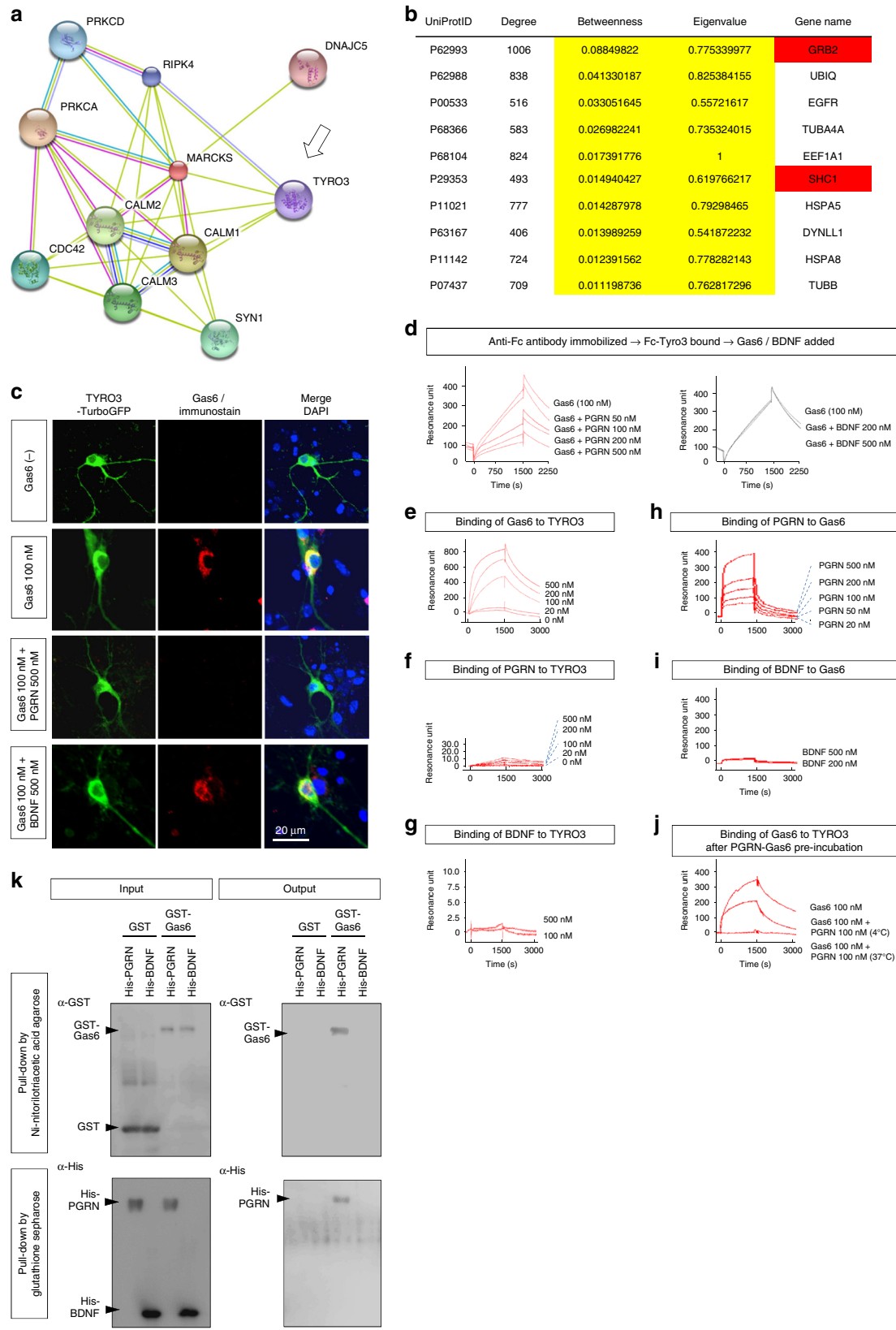

activation of phospholipase C (PLC)[33]. Moreover, a recent report based on whole-genome sequencing of 1345 familial late-onset AD (LOAD) patients revealed that PKC mutations with increased catalytic activity are associated with disease onset[34].

According to the hypothesis, we formulated based on the signaling map (Supplementary Fig. 2D), a candidate receptor for PGRN should engage in cross-talk with neighboring receptors, such as Toll-like receptors, at the level of their downstream signals, and should activate upstream molecules of the B-Raf/MEK/ERK pathway such as SHC and GRB. Based on informatics analysis, Tyro3 satisfied all of these conditions: Grb2 and Shc1 were listed among top 10 genes when candidate molecules to connect TLR4 and Tyro3 pathways were expected by calculating centrality scores (betweenness and eigenvalue) in the PPI network composed of the 3-hop genes from TLR4 and Tyro3 (Fig. 2b). Therefore, we selected Tyro3 as the best candidate for a novel PGRN receptor.

We then asked whether Tyro3 is a receptor for PGRN, and if so, whether PGRN can block the signal from Tyro3 to the MAPK pathway. PGRN inhibited the interaction between Gas6 and Tyro3-TurboGFP expressed on the plasma membrane of primary cortical neurons from E15 embryos (Fig. 2c). Surface Plasmon Resonance (SPR) analysis confirmed that PGRN inhibited binding of Gas6 to Tyro3 in a concentration-dependent manner (Fig. 2d, left panel). No such inhibition was observed with BDNF, used as a negative control (Fig. 2d, right panel). In contrast to the binding of Gas6 to Tyro3 (Fig. 2e), PGRN did not bind to Tyro3 by itself (Fig. 2f), nor did BDNF interact with Tyro3 (Fig. 2g). PGRN did bind to Gas6 (Fig. 2h), whereas BDNF did not (Fig. 2i). Moreover, pre-incubation with PGRN, especially at 37 ° C, prevented binding of Gas6 to Tyro3 (Fig. 2j).

**PGRN inhibits Gas6–Tyro3 signaling**. The pathway from Tyro3 leads to activation of B-Raf and MAPK signaling via Shc and Grb (Supplementary Fig. 2D). In addition, this pathway activates PKC via DAG synthesis and PLC[33], or via an alternative pathway from Shc to PKC[35]. Consistent with the inhibitory effect of PGRN on the interaction between Gas6 and Tyro3 (Fig. 2), western blot analyses revealed in vivo activation of Tyro3 (Fig. 3a) and phosphorylation of Shc and PLCγ (Fig. 3b, c) in whole cerebral cortex tissue of PGRN-KI mice.

To confirm this observation, we investigated whether PGRN affects downstream signals of Tyro3 in mouse primary cortical neurons (Fig. 3d–f). As expected, addition of Gas6 to primary cultures induced phosphorylation of Tyro3, and simultaneous addition of PGRN inhibited Tyro3 phosphorylation in a concentration-dependent manner (Fig. 3d). PGRN also suppressed Gas6-induced phosphorylation of Shc and B-Raf in a concentration-dependent manner (Fig. 3e). Gas6-induced phosphorylation of PLCγ and simultaneous addition of PGRN inhibited PLCγ phosphorylation in a concentration-dependent manner (Fig. 3f). Accordingly, PGRN inhibited PKC phosphorylation downstream of Gas6 signaling in a concentration-

dependent manner (Fig. 3f). The suppression was most remarkable for PKCα, and the effect was less obvious for PKCγ and PKCδ (Fig. 3f).

Moreover, siRNA-mediated knockdown (KD) of Tyro3 suppressed Gas6-induced activation and phosphorylation of Tyro3 by itself (i.e., autophosphorylation), Shc, PLCγ, and tau at Ser203 and Thr220 to the same extent as the KD suppressed phosphorylation by PGRN (Fig. 4a–f), confirming that the suppressive effect of PGRN on Gas6 was mediated by Tyro3.

**Gas6–Tyro3 signals are responsible for pSer203-tau**. The differences between the outcomes of tau phosphorylation at Ser203 and Thr220 in response to the B-Raf inhibitor suggested differences in the signaling pathways downstream of these modifications. To investigate this issue, we first surveyed candidate kinases responsible for tau phosphorylation at Ser203 and Thr220 using the KISS approach[36–38] and associated database KANPHOS (https://kanphos.neuroinf.jp)[39]. PKA, CaMKII, and PKCα were predicted to be responsible for tau phosphorylation at Ser203/214. Although we did not obtain results for tau phosphorylated at Thr220/231 in the same screen (our unpublished observation), PKA, CaMKII, Cdk5, and GSK3b are also involved in phosphorylation at this site[40].

In in vitro phosphorylation assays with GST-tau and GST-kinase expressed in baculovirus, PKA, RhoK, CaMKII, and PKCα strongly phosphorylated tau at Ser203/214 (upper panels in Fig. 5a). Moreover, an in vivo assay of COS7 cells expressing GFP-tau and GST-kinase indicated that all of these candidate kinases, including ERK2/MEK1, could cause phosphorylation of tau at Ser203/214, either directly or indirectly (upper panels in Fig. 5b). On the other hand, Cdk5/p35 and GSK3β could directly phosphorylate tau at Thr220/231 in vitro (upper panels in Fig. 5a). All of the kinases examined, including ERK2/MEK1, also induced tau phosphorylation at Thr220/231 in vivo, probably by activating the directly responsible kinases (upper panels in Fig. 5b). These results indicate that different groups of kinases are responsible for direct phosphorylation of tau at Ser203 and Thr220, whereas multiple kinases indirectly trigger phosphorylation of tau in vivo. We also subjected human Ser214Ala (corresponding to mouse Ser203) and Thr231Ala (corresponding to mouse Thr220) mutants to the in vitro and in vivo assays. The mutant proteins were not phosphorylated by the candidate kinases (middle and lower panels in Fig. 5a, b).

Analysis of comprehensive phosphoproteome data (Supplementary Fig. 3) confirmed activation of the PKC signaling pathway, but not the PKA, RhoK, or CaMKII pathways, in PGRN-KI mice at 4 weeks of age (Supplementary Fig. 2C), supporting the idea that PKCα is the most plausible candidate for the kinase that directly phosphorylates tau at Ser203/214. However, in light of the results of the in vivo phosphorylation assay (Fig. 5b), it remains possible that other kinases, including ERK/MAPK, indirectly cause phosphorylation of tau at Ser203/214.

---

**Fig. 2** PGRN prevents Gas6 interaction with Tyro3. **a** PPI network of MARCKS retrieved from String (http://string91.embl.de/, http://string-db.org/) suggests that Tyro3 is a candidate receptor related to MARCKS. **b** Downstream molecules of TNFR and Tyro3 were retrieved from PPI database of the Genome Network Platform of the National Institute of Genetics (http://genomenetwork.nig.ac.jp/index_e.html), from which Top100 molecules with highest centrality scores (betweenness and eigenvalue) were listed. 10 molecules are shared by Top100 lists of TNFR and Tyro3, in which Shc and Grb2 are identified as critical signal mediators. **c** Interaction of Gas6 with the plasma membrane in primary cortical neurons expressing Tyro3-TurboGFP, and inhibition of this interaction by PGRN. **d** SPR analysis revealed Gas6 binding to Tyro3 and the inhibition of this interaction by PGRN (left panel). BDNF did not inhibit binding of Gas6 to Tyro3 (right panel). **e, f, g** Affinities of Gas6, PGRN, and BDNF to Tyro3 were evaluated by SPR. Gas6, but not PGRN or BDNF, bound Tyro3 with high affinity. **h** PGRN bound Gas6 at high affinity. **i** BDNF did not bind to Gas6. **j** Pre-incubation of Gas6 with PGRN decreased the affinity of Gas6 for Tyro3. **k** Pull-down assay revealed direct interaction between PGRN and Gas6. His-tagged proteins were pulled down by Ni-NTA-agarose, and GST-tagged proteins were pulled down by glutathione sepharose

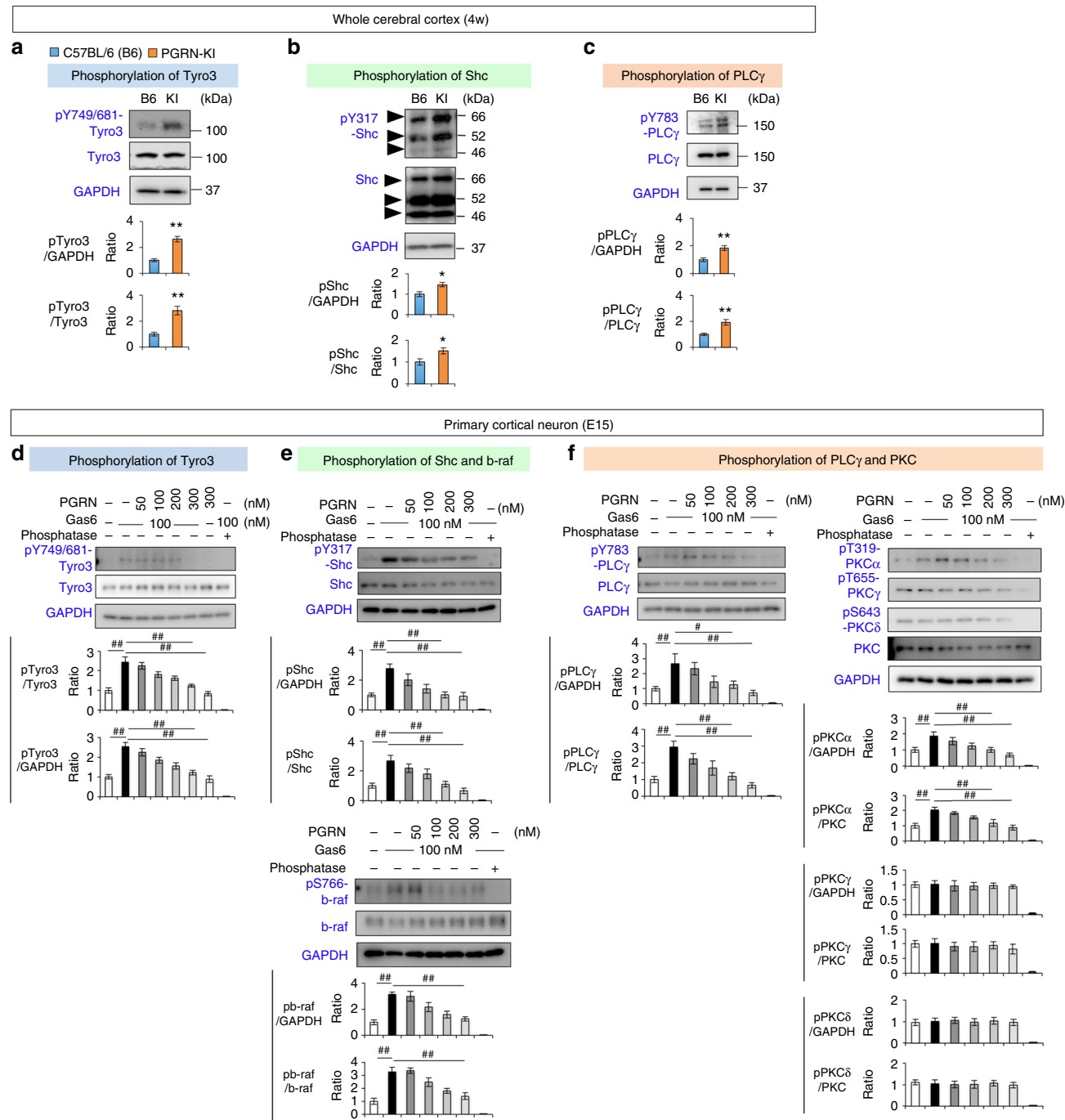

**Fig. 3** PGRN inhibits downstream signals of Tyro3. **a** Western blot analyses show elevated phosphorylation of Tyro3 in the whole cerebral cortex sample prepared from PGRN-KI mice. Lower graphs show quantitation of pTyro3/GAPDH and pTyro3/Tyro3 ratios. *$p < 0.05$; **$p < 0.01$ ($N = 6$, Student's $t$-test). Averages and s.e.m. are shown. **b** Western blot analyses show elevated phosphorylation of Shc was elevated in cerebral cortex of PGRN-KI mice. *$p < 0.05$; **$p < 0.01$ ($N = 6$, Student's $t$-test). Averages and s.e.m. are shown. **c** Western blot analyses show elevated phosphorylation of PLCγ is increased in cerebral cortex of PGRN-KI mice. *$p < 0.05$; **$p < 0.01$ ($N = 6$, Student's $t$-test). Averages and s.e.m. are shown. **d** Western blot analyses of Tyro3. Addition of Gas6 to culture medium increased phosphorylation of Tyro3 in cortical neurons (E15) at day 7 in primary culture, whereas co-addition of PGRN suppressed Gas6-induced phosphorylation of Tyro3. ##$p < 0.01$ ($N = 4$, Dunnett's test). Averages and s.e.m. are shown. **e** Western blot analyses of Shc and B-Raf. Gas6 increased phosphorylation of Shc and B-Raf in mouse primary cortical neurons at day 7, whereas PGRN suppressed the activation induced by Gas6. ##$p < 0.01$ ($N = 4$, Dunnett's test). Averages and s.e.m. are shown. **f** Western blot analyses of PLCγ and PKCα. Gas6 activated PLCγ and PKCα in mouse primary cortical neurons at day 7, whereas PGRN suppressed Gas6-induced activation of PKCα. PKCγ and PKCδ were not remarkably affected by Gas6. #$p < 0.05$; ##$p<0.01$ ($N = 4$, Dunnett's test). Averages and s.e.m. are shown

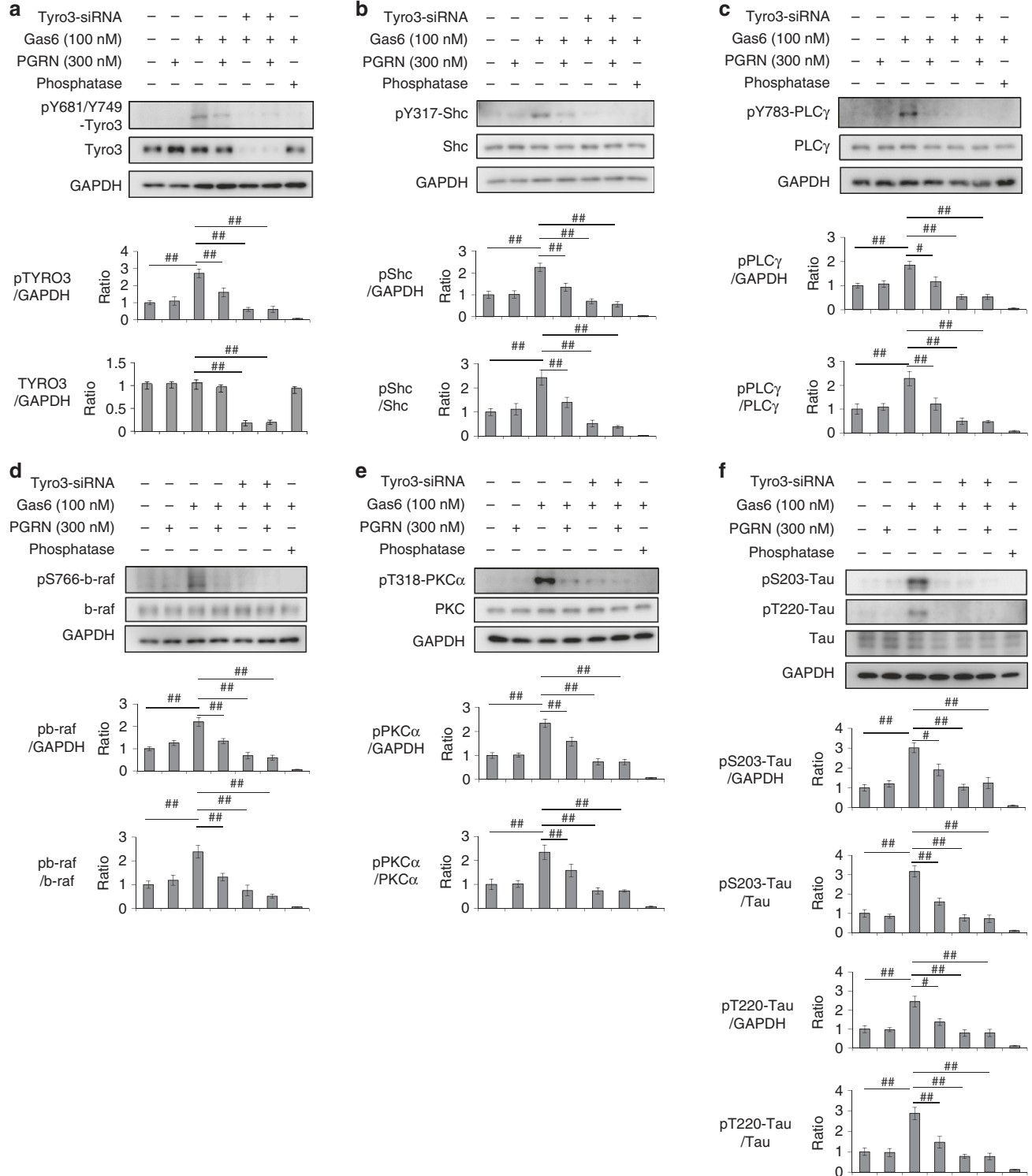

**Fig. 4** Tyro3 mediates Gas6-induced signal to tau phosphorylation. Western blot analyses of the effect of siRNA-mediated knockdown of Tyro3 on phosphorylation of Tyro3 (**a**), Shc (**b**), PLCγ (**c**), B-Raf (**d**), PKCα (**e**), and tau (**f**). SiRNA-mediated knockdown of Tyro3 exerted suppressive effects on activation of Shc (**b**) and PLCγ (**c**), as well as on phosphorylation of tau at Ser203 and Thr220 (**e**). $^{#}p < 0.05$; $^{##}p < 0.01$ ($N = 5$, Tukey's HSD test). Averages and s.e.m. are shown. Phosphatase treatment of the samples substantially decreased the western blot band intensities of phosphorylated Tyro3 (**a**), Shc (**b**), PLCγ (**c**) B-Raf (**d**), PKCα (**e**), and Tau (**f**). p-values are shown in Supplementary Data 1

**Inhibition of Gas6–Tyro3 signals recues cognitive impairment.**
Next, we asked whether inhibition of PKCα by Gö6976 or inhibition of B-Raf by vemurafenib (PLX4032) would improve the behavioral phenotypes of PGRN-KI mice (Figs. 6 and 7). Vemurafenib, an inhibitor of B-Raf, is used as an anti-cancer drug

to target melanoma with B-Raf mutations, although the mechanisms underlying the drug's effects remain controversial. The original study by Tsai et al.[41] reported that, at 100 nM, vemurafenib suppressed phosphorylation of ERK in a melanoma cell line harboring the B-Raf V600E mutation, but also showed

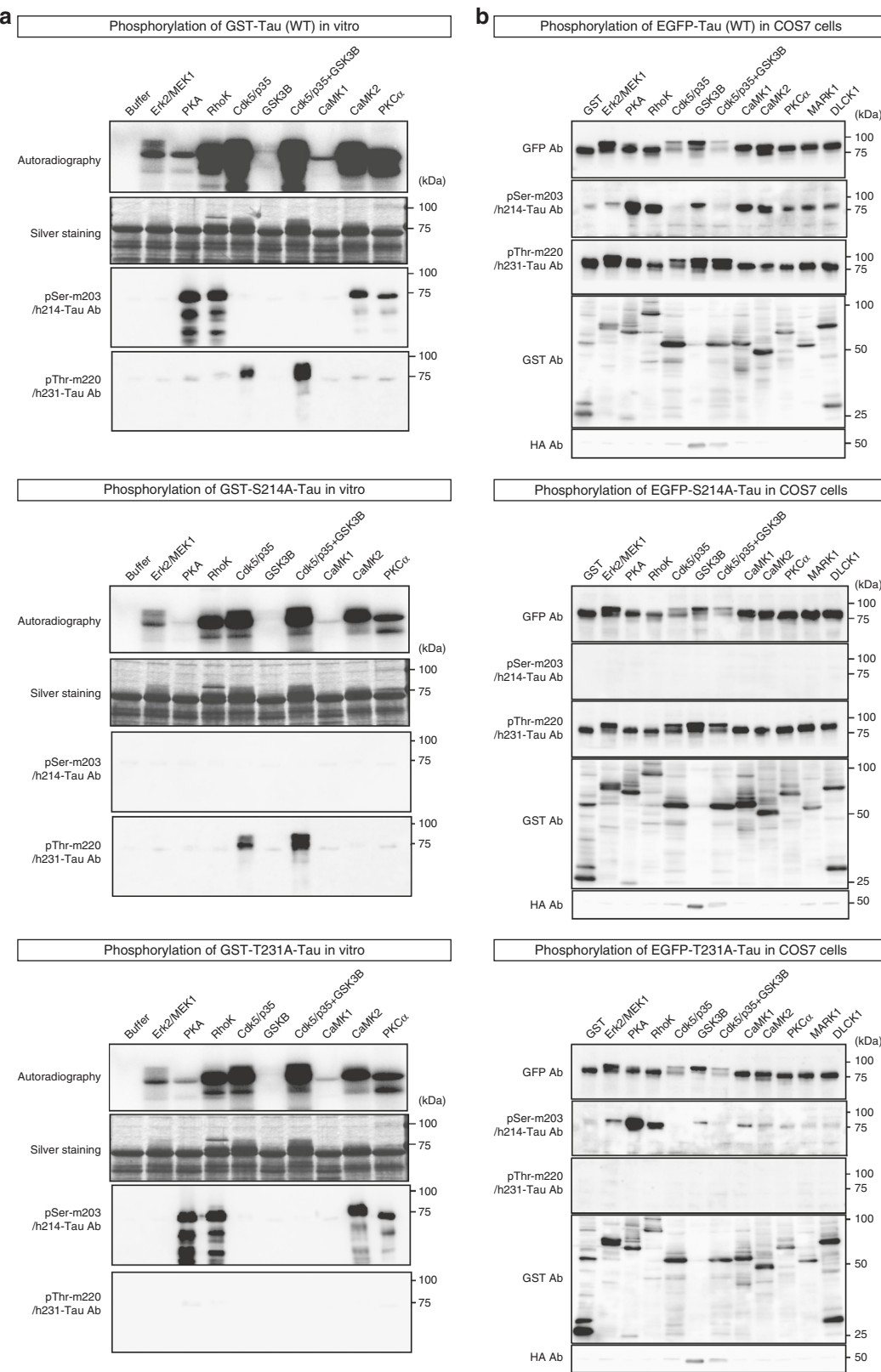

that at concentrations over 1 µM it suppressed phosphorylation of ERK in another melanoma cell line with wild-type B-Raf. A subsequent paper by Bollag et al.[42] reported that the IC50 of vemurafenib was 31 nM for B-Raf-V600E and 100 nM for wild-type B-Raf. Structure biology analysis also showed that Vemurafenib binds to the hinge region of b-raf with kinase activity, for which V600E mutation is not essential[43]. Therefore, although vemurafenib and its derivatives[42] are sometimes erroneously believed to specifically suppress the B-Raf V600E mutant, they also suppress wild-type B-Raf at relatively low concentrations.

Accordingly, to suppress wild-type B-Raf, we administered an oral dose of 32 mg/kg of BW/day vemurafenib; this dose was derived from the dose approved by the FDA for use in melanoma patients (https://www.accessdata.fda.gov/drugsatfda_docs/label/2011/202429s000lbl.pdf). PGRN-KI mice were fed vemurafenib or an equivalent volume of PBS every day from 6 to 12 weeks of age (Fig. 6a). Western blot of whole cerebral cortex revealed that vemurafenib suppressed active wild-type B-Raf phosphorylated at Ser766 (Fig. 6b), even though the dose was intended to suppress B-Raf-V600E. Other molecules in the MAPK signaling pathway, such as MEK1 and ERK1, were also suppressed by vemurafenib (Fig. 6c, d). Moreover, vemurafenib reduced the level of Ser203-phosphorylated tau, but not the Thr220-phosphorylated form (Fig. 6e). Morris water maze and fear-conditioning tests revealed that vemurafenib-treated mice recovered memory and cognitive function (Fig. 6f, g).

Similarly, we examined the in vivo effect of Gö6976, a PKC inhibitor (4.4 µM, 0.15 µl/h, intrathecal administration), in mice 10 to 12 weeks of age (Fig. 7a). Gö6976 suppressed phosphorylation of PKCα and tau at Ser203 (Fig. 7b, c) and rescued the cognitive impairment of PGRN-KI mice, as determined by the Morris water maze and fear-conditioning tests (Fig. 7d, e). Together, the therapeutic effects of PKC and B-Raf inhibitors support the idea that upregulation of Gas6–Tyro3 signaling leads to cognitive phenotypes in vivo.

Inhibition of neither B-Raf nor PKC affected aggregation or phosphorylation of TDP43 in PGRN-KI mice at 12 weeks of age (Supplementary Fig. 6), suggesting that the change in tau phosphorylation precedes TDP43 pathology.

**B-Raf activation and suppression of PGRN-KI mice in vivo.** To rule out the possibility that phosphorylation and suppression of the B-Raf pathway in mouse brain were postmortem artifacts, we performed in vivo imaging of ERK. To this end, we injected lentiviral vectors expressing ERK/MAPK- or JNK-substrate-CFP/YFP fusion proteins (lenti-EKAREVnes, lenti-JNKAREVnls) into the cerebral cortex of PGRN-KI mice (Supplementary Fig. 4A), and then measured kinase activities based on intra-molecular fluorescence resonance energy transfer (FRET) from Turquoise-GL(CFP) to Ypet(YFP)[44]. The results supported the idea that MAPK/ERK was indeed activated in vivo in the cortex of PGRN-KI mice, and that vemurafenib suppressed activation of the MAPK kinase pathway in these animals (Supplementary Fig. 4B,

C). No such changes in FRET were observed with JNK, used as a negative control (Supplementary Fig. 4B, C). We could not detect PKC activation in vivo because no appropriate lentiviral vector was available (data not shown).

To evaluate the extent and specificity of vemurafenib's effects, we next performed phosphoproteome analyses on whole cerebral cortex tissues from vemurafenib-treated and non-treated PGRN-KI mice (Supplementary Fig. 5). The samples were prepared immediately after the behavioral tests (Fig. 6). Specifically, we evaluated the phosphorylation states of B-Raf downstream molecules, identified from the integrated PPI database (http://genomenetwork.nig.ac.jp/index_e.html)[31] as molecules connected to B-Raf by two or fewer degrees of separation. In the diagram in Supplementary Fig. 5A, green indicates proteins whose phosphorylation was suppressed with a $q$-value < 0.05, white indicates no statistically significant change, and gray indicates proteins detected by mass spectrometry in neither vemurafenib-treated nor non-treated PGRN-KI mice with > 95% confidence. Diagonal and circular nodes indicate kinase and non-kinase proteins, respectively. Ultimately, five kinases downstream of B-Raf were detected by mass spectrometry, of which four were suppressed by vemurafenib. A total of 31 non-kinase proteins were detected downstream of B-Raf, of which 27 were suppressed by vemurafenib (Supplementary Fig. 5A).

To assess the specificity of vemurafenib's effects, we calculated the ratios of numbers of the suppressed phosphorylated proteins or phosphorylation sites compared to the numbers of detected phosphorylated proteins or phosphorylation sites, respectively, in the B-Raf–downstream and B-Raf–independent categories (Supplementary Fig. 5B). The ratios of suppression were significantly higher in the B-Raf–downstream category (Fisher's exact test, $p =$ 5.69E−05 for proteins and $p = 1.64$E−12 for phosphorylation sites). Although the effect of vemurafenib was not perfectly selective, the drug was sufficient to suppress the B-Raf downstream signaling pathway. Hence, in the following experiments, we further investigated the direct roles of B-Raf and tau in impairment of cognition and memory.

**pSer203-tau mislocalization leads to post-synaptic pathology.** To determine how signaling leading to tau phosphorylation impairs cognition and memory, we investigated the post-synaptic structure of dendritic spines, assuming that phosphorylated tau plays a similar pathological role to that proposed for AD pathology[45–48]. To this end, we performed in vivo imaging of dendritic spines in layer 1 of frontal cortex M2 by two-photon microscopy. We injected EGFP-expressing AAV into RSD adjacent to M2 of non-treated PGRN-KI and C57BL/6J mice at 6 weeks of age (Fig. 8a). In vivo imaging at 8 weeks of age clearly revealed significant reduction of spine density in PGRN-KI mice (Fig. 8b), whereas other spine parameters such as length, diameter, and volume did not differ from those in C57BL/6J mice (Fig. 8b). To evaluate spine dynamism, we performed serial observations at three time points, and observed an increase in

**Fig. 5** Identification of kinases responsible for tau phosphorylation. **a** In vitro phosphorylation assay with GST-tau and GST-kinase expressed in baculovirus. Autoradiography indicated that GST-tau was phosphorylated by all the kinases and labeled with γ-$^{32}$P-ATP during the phosphorylation reaction. Specific antibodies against mouse Ser203/human Ser214 or mouse Thr220/human Thr231 detected the kinases responsible for phosphorylation at each site. Results for GST-human tau (wild type), GST-human tau S214A mutant, and GST-human tau Thr231Ala mutant are shown in the upper, middle, and lower panels. Human tau Ser214 and Thr231 correspond to mouse Ser203 and Thr220, respectively. Thus, the middle and lower panels show negative controls. **b** In vivo assay of COS7 cells expressing GFP-tau and kinases tagged with GST or HA. Expression levels of tau proteins and kinases were confirmed with anti-GFP or anti-GST/HA antibody, respectively. Western blots with specific antibodies against mouse Ser203/human Ser214 or mouse Thr220/human Thr231 revealed that all kinases could phosphorylate tau at each site. Results for EGFP-human tau (wild type), EGFP-human Ser214Ala mutant, and EGFP-human tau Thr231Ala mutant are shown in the upper, middle, and lower panels

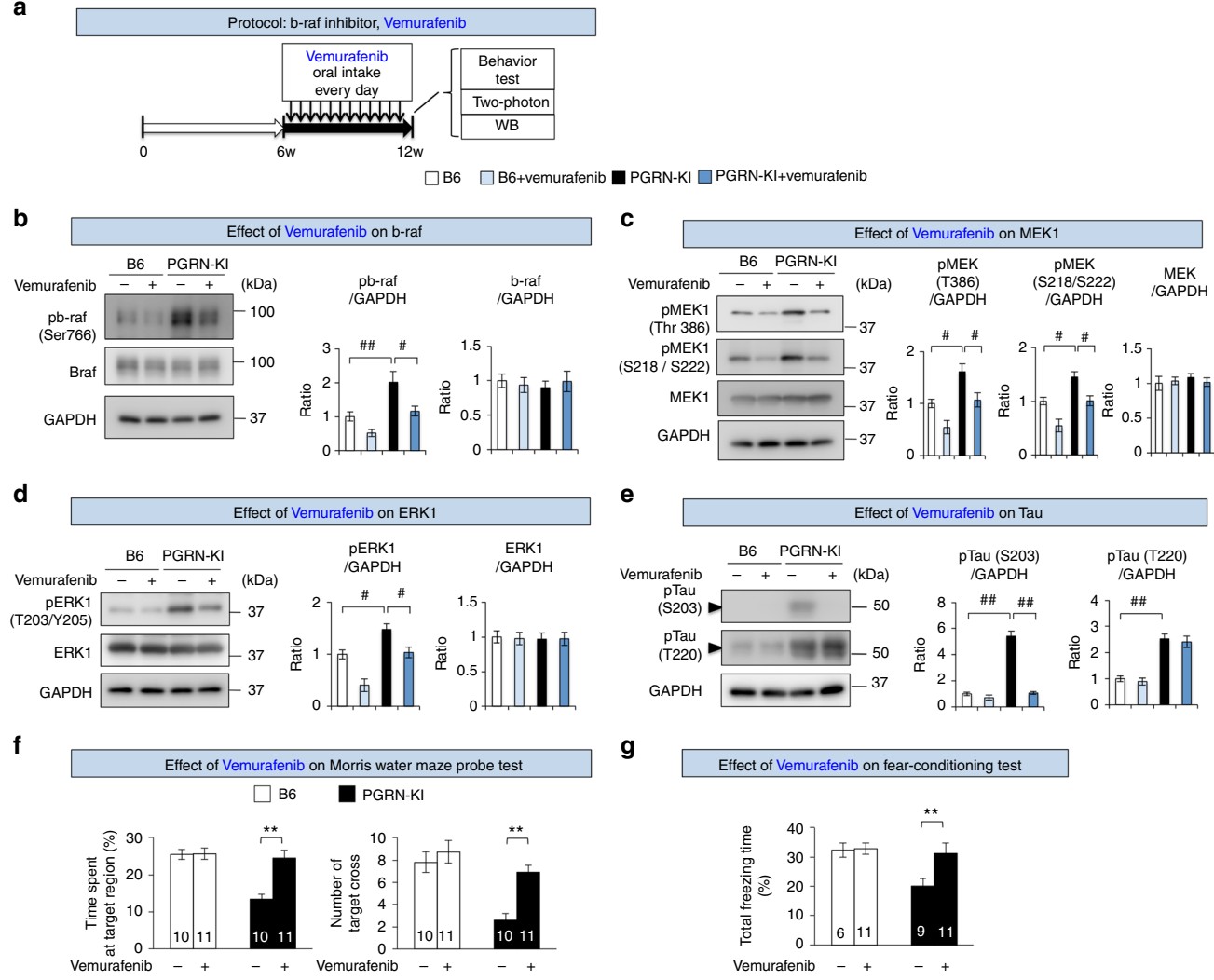

**Fig. 6** Vemurafenib rescues phosphorylation signals and phenotypes in PGRN-KI mice. **a** Experimental protocol for administration of B-Raf inhibitor (vemurafenib). The mice were fed vemurafenib (32 mg/kg of BW/day) or mock from 6 to 12 weeks of age. Behavioral tests and western blots were performed at 12 weeks. **b** In vivo effect of vemurafenib on B-Raf phosphorylation in cortical tissues of PGRN-KI mice. #$p < 0.05$; ##$p<0.01$ ($N = 6$, Tukey's HSD test). Averages and s.e.m. are shown. **c** In vivo effect of vemurafenib on MEK phosphorylation in cortical tissues of PGRN-KI mice. #$p < 0.05$ ($N = 6$, Tukey's HSD test). Averages and s.e.m. are shown. **d** In vivo effect of vemurafenib on ERK phosphorylation in cortical tissues of PGRN-KI mice. #$p < 0.05$ ($N = 6$, Tukey's HSD test). Averages and s.e.m. are shown. **e** In vivo effect of vemurafenib on phosphorylation of tau in cortical tissues of PGRN-KI mice. Right panels show quantitation of western blot band intensities. ##$p < 0.01$ ($N = 6$, Tukey's HSD test). Averages and s.e.m. are shown. **f** In vivo effect of vemurafenib on two parameters (% time spent at target region and number of target crosses) in the Morris water maze test. Numbers of mice are shown in the graph. **$p < 0.01$ (Student's $t$-test). Averages and s.e.m. are shown. **g** Effect of vemurafenib on % total freezing time in the fear-conditioning test. Numbers of mice are shown in the graph. $p$-values were determined by Student's $t$-test. Averages and s.e.m. are shown

spine elimination and a decrease in the abundance of stable spines over 24 h (Fig. 8c).

Next, we injected KD vectors expressing sh/siRNA targeting B-Raf, PKCα, Gas6, or Tyro3 into 8-week-old mice and extended the analyses for another 4 weeks (Fig. 8a). In vivo imaging at 12 weeks revealed recovery of spine density in KD vector-treated PGRN-KI mice, and confirmed that inhibition of B-Raf signaling molecules rescued pre-synapse structure (Fig. 8d–h). In addition, the imaging showed that administration of vemurafenib and Gö6976 rescued the reduction in the number of spines in mutant PGRN-KI mice (Fig. 8i, j).

Immunohistochemistry using antibodies against tau phosphorylated at Ser203 or Thr220 (Fig. 9a) yielded distinct patterns of staining in cerebral cortex of PGRN-KI mice. Anti-Ser203 antibody stained neuropils and revealed mislocalization of

Ser203-phosphorylated tau to dendritic spines co-stained with PSD95 (Fig. 9a). By contrast, anti-Thr220 antibody stained cytoplasm and neurites but not PSD95-positive spines (Fig. 9a). The number of PSD95-positive dots was reduced in PGRN-KI mice (Fig. 9a, b). In addition, the number of PSD95 dots co-stained with Ser203-phosphorylated tau, but not with Thr220-phosphorylated tau, was elevated in PGRN-KI mice (Fig. 9a, b). As expected, KD of B-Raf signaling molecules prevented the decrease in the number of PSD95-positive dots and the increase in co-localization between PSD95 and Ser203-phosphorylated tau (Fig. 9a, b). Consistent with this, vemurafenib and Gö6976 also rescued the mislocalization of pSer203-tau to spines and normalized the number of PSD-positive dots (Fig. 9a, b).

Collectively, these results supported the idea that B-Raf, PKC, and tau are directly involved in mislocalization of pSer203-tau

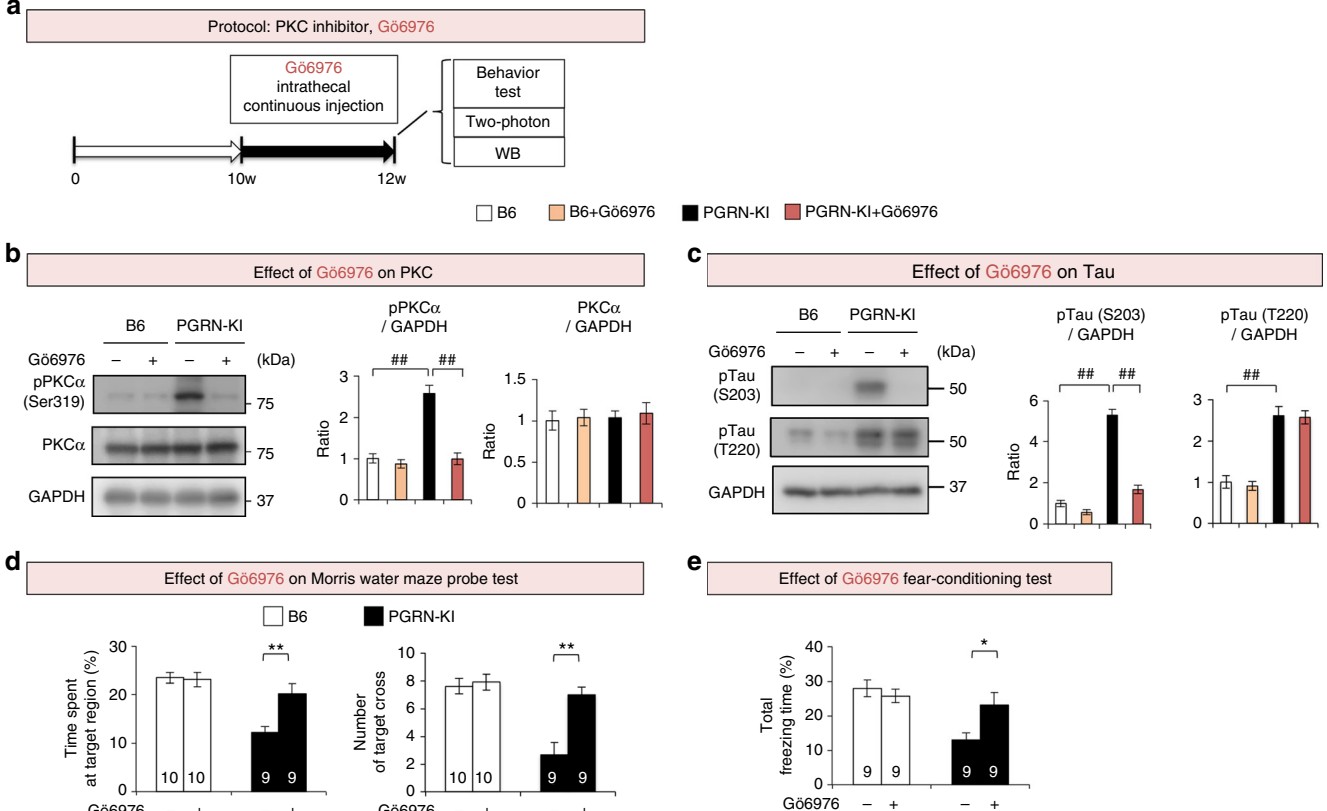

**Fig. 7** Gö6976 rescues phosphorylation signals and phenotypes in PGRN-KI mice. **a** Experimental protocol for PKC inhibitor (Gö6976). Gö6976 (4.4 μM) or PBS was injected via osmotic pump into the subarachnoid space of PGRN-KI and C57BL/6J mice from 10 to 12 weeks of age, and behavioral tests and western blots were performed at 12 weeks. **b** In vivo effect of Gö6976 on PKCα in cortical tissues of PGRN-KI mice. ##$p < 0.01$ ($N = 6$, Tukey's HSD test). Averages and s.e.m. are shown. **c** In vivo effect of Gö6976 on phosphorylation of tau in cortical tissues of PGRN-KI mice. Right panels show quantitation of western blot band intensities. ##$p < 0.01$ ($N = 6$, Tukey's HSD test). Averages and s.e.m. are shown. **d** In vivo effect of Gö6976 on two parameters (% time spent at target region and number of target crosses) in Morris water maze test. $p$-values were determined by Student's $t$-test ($N = 10$). Averages and s.e.m. are shown. **e** Effect of Gö6976 on % total freezing time in the fear-conditioning test. $p$-values were determined by Student's $t$-test ($N = 9$). Averages and s. e.m. are shown. $p$-values are shown in Supplementary Data 1

and the resultant decrease in the number of dendritic spines. Together with previous observations made in relation to AD[46,49], our results suggest that mislocalization of phosphorylated tau to spines is a common mechanism underlying synapse pathologies across multiple types of dementia.

**Behavioral rescue by KD of tau and signal molecules**. To test the direct effects of tau, B-Raf, PKCα, Tyro3, and Gas6 on the behavioral phenotypes of PGRN-KI mice, we evaluated behavioral recovery in PGRN-KI mice injected with sh/siRNA KD vectors targeting these proteins, following the protocol described above (Fig. 8a). Consistent with the observed recovery of spine density, PSD95-positive spots, and pSer203-tau mislocalization, the KD vectors rescued memory impairment (as determined by the Morris water maze test, Fig. 9c) and impaired cognitive function (as determined by the fear-conditioning test, Fig. 9d) in PGRN-KI mice. The recovery was not strengthened by double treatment with tau-KD and Vemurafenib or Gö6976 (Fig. 9e), supporting the idea that both treatments target the same signaling pathway. Together, these results indicate that tau and its upstream signal molecules are directly involved in the post-synaptic changes and the resultant behavioral phenotypes of PGRN-KI mice. We confirmed the effect of shRNA lentiviral vectors on their target proteins by western blot analyses of cerebral cortex tissues (Fig. 9f).

**pSer214-tau mislocalization in sporadic human FTLD-TDP**. Finally, we investigated whether spine pathology induced by pSer203-tau occurs in human FTLD-TDP. Immunohistochemistry of brain tissue from human autopsies ($N = 5$) revealed that sporadic human FTLD cases with TDP43 aggregation exhibit similar mislocalization of Ser214-phosphorylated tau to PSD95-positive post-synaptic spines in frontal and temporal cortex (Fig. 10a), corresponding to the mislocalization of Ser203-phosphorylated tau to PSD95-positive spines in mice (Fig. 9). Pathological diagnoses of FTLD-TDP in autopsied subjects were confirmed by immunohistochemistry using anti-phospho-TDP43 antibody (Fig. 10b).

Assuming that spine pathology arising at the earliest time points continues during the progression of disease, this observation strongly suggested that the mechanism revealed in mice is applicable to the molecular pathology of human FTLD.

**Discussion**
Protein phosphorylation has been intensively investigated in AD, specifically in the context of tau aggregation. Multiple reports suggest that tau phosphorylation promotes formation of aggregates, including paired helical filaments (PHFs)[50], although postmortem artifacts have been proposed to be responsible for the tau phosphorylation observed in postmortem AD brains.

Physiologically, phosphorylation reduces the affinity of tau to α-tubulin as MAP[51,52] and destabilizes microtubules[53]. Independently of tau aggregation, mislocalization of phosphorylated tau increases Aβ toxicity, impairs synaptic function, and impairs memory formation in AD model mice[45–48].

Several studies reported activation of serine/threonine kinases, including GSK3beta, JNK, PKA, Cdk5, and casein kinase II, in AD, and demonstrated the involvement of these factors in tau phosphorylation[54–60]. However, the overall scheme of the phosphorylation signaling network in AD remains unclear. Previously, we performed comprehensive phosphoproteome analysis

(60,000–100,000 peptides, >95% reliability) of brain samples from four AD mouse models and human AD patients[31]. The results revealed a dense core network composed of PKC-phosphorylated proteins that formed prior to amyloid accumulation. Notably in this regard, an independent study that performed whole-genome sequencing of 1,345 patients from 410 families with LOAD detected five families harboring hyperactivated variants of PKCα[34]. Together, the results of these two independent studies, which applied different comprehensive approaches at the gene and protein levels, suggest a role for a novel signaling pathway in AD pathogenesis.

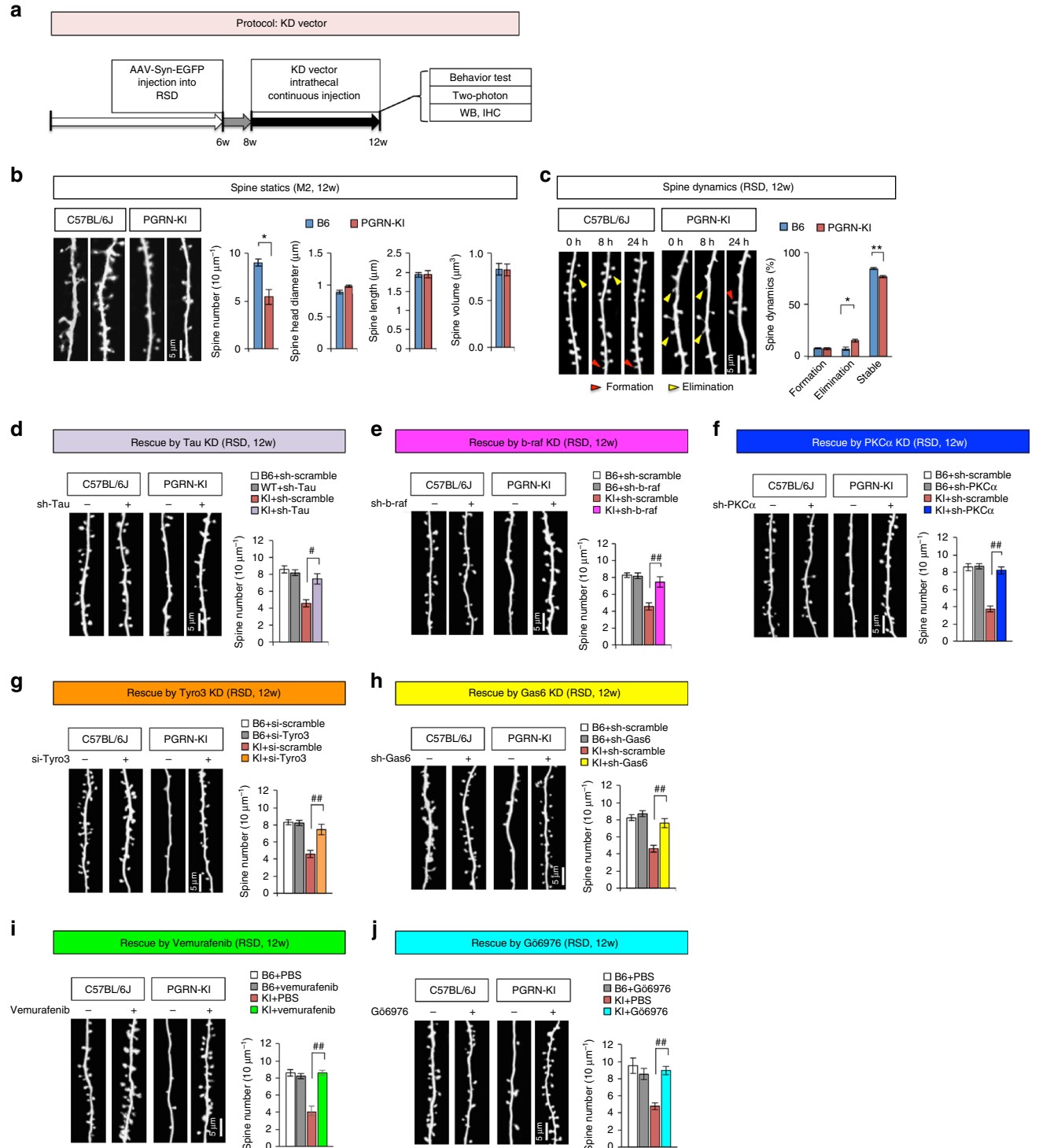

Previously, however, no comprehensive study has aimed to elucidate the core phosphorylation signal network in FTLD at a deep level. Because phosphoproteins in postmortem brains change rapidly[61], it is essential to perform proteome analysis in animal models rather than human brains obtained from autopsies, and human data should be used only as a reference. In this regard, our analysis with a knock-in mouse model that mimics human pathology more closely than transgenic or knockout models would be important.

FTLD is a category of dementia that predominantly affects the frontal and temporal lobes. The neural circuits corresponding to lobar atrophy are obviously distinct from those involved in other types of dementia, such as AD. Because the neuronal aggregates are not homogeneous, FTLD is subcategorized based on the aggregates components as FTLD-tau, FTLD-TDP, FTLD-FUS, and FTLD-UPS[29].

Notably in this regard, the first conclusion of the present study was that tau phosphorylation signaling is activated in the tau-negative form of PGRN-linked FTLD (Supplementary Fig. 7). Based on the chronological relationship between the two events, TDP43 aggregation cannot act upstream of tau phosphorylation: phosphorylated tau in the brain of PGRN-KI mice started at 4 weeks of age, whereas TDP43/ubiquitin staining was not detectable until 24 weeks. Several alternative hypotheses could explain the relationship between tau and TDP43. First, the kinases responsible for tau phosphorylation at Ser203 might act upstream and affect TDP43 aggregation. However, this is not plausible because suppression of MAPK or PKC did not affect TDP43 aggregation in vivo (Supplementary Fig. 6). Second, the signal responsible for TDP43 aggregation might also promote phosphorylation of tau in parallel. Although the signaling pathways that promote TDP43 aggregation[62–67] are distinct from the PKC or MAPK signal pathways leading to tau phosphorylation at Ser203, cross-talk between the pathways and resultant indirect effect remain possible as suggested in our experiment.

In summary, we speculate that the signals initiated by disinhibited Tyro3 trigger activation of PKC and induce tau mislocalization. In parallel, these signals might affect other kinases that modulate TDP43 aggregation through a gel–sol phase transition[19–21]. This hypothesis explains the multiple phases of pathology, which progress from functional disturbances of synapses to aggregation, and ultimately to neuronal death.

The second critical finding of this study is that PGRN is a negative modulator of the Gas6–Tyro3 interaction. Because our data were not consistent with the idea that TNFR signaling is affected in PGRN-KI mice, we searched for a new receptor of PGRN. The top candidate was Tyro3, and our investigations of this factor revealed that PGRN inhibited binding of Gas6 to Tyro3 (Figs. 3 and 4). Specifically, PGRN blocked activation of Tyro3 and suppressed activation of its downstream MAPK signaling pathway (Figs. 3, 4, and 5). Conversely, deficiency of PGRN de-repressed Tyro3 and induced abnormal activation of MAPK signaling. These findings provide new insight into the pathomechanism of FTLD.

The biological importance of Gas6–Tyro3 ligand/receptor interaction remains incompletely understood, and only a few studies have implicated Gas6–Tyro3 signaling in neurodegenerative diseases. Based on experiments in primary neuron culture, one report preliminarily suggested that Gas6 suppresses amyloid-β–induced apoptosis[68], and another showed that production of amyloid-β is decreased by overexpression of Tyro3 in HEK293 cells[69]. However, the former study lacked animal- or molecular-based evidence, and the latter reported the contradictory finding that Gas6 inhibits the effect of Tyro3.

TYRO3 belongs to the TAM family, whose members are receptors for Gas6/PROS1 on the surface of phosphatidylserine-presenting apoptotic cells or are secreted from apoptotic cells and platelets[32,33]. The TAM receptors Tyro3, AXL, and MERTK are oncogenic drivers that promote survival, chemoresistance, and motility of cancer cells[32]. Autophosphorylation of Tyro3 following ligand binding induces activation of the MAPK pathway through Shc and Grb, consistent with the pro-proliferative role of Tyro3[70–72], and activates PKC via PLC[33]. TAM family receptors promote or suppress inflammation in a cell type–dependent manner. $Tyro3^{-/-} Axl^{-/-} Mertk^{-/-}$ triple-KO mice exhibit loss of neurons in old age[73], but no phenotype of Tyro3 overactivation has been reported, and it is not possible to confidently speculate whether this protein promotes FTLD or other neurodegenerative diseases.

Third, among two or three candidate phosphorylation sites on tau, our results specifically connect synapse pathology to tau phosphorylation at Ser203 (which corresponds to human Ser214) (Figs. 8, 9, and 10). B-Raf inhibitor suppressed tau phosphorylation at Ser203, but not at Thr220 (Fig. 6e). Tau phosphorylated

**Fig. 8** Suppression of Tyro3 signal rescues decrease in spine abundance and tau mislocalization in mutant PGRN-KI mice. **a** Experimental protocol for knockdown vectors. AAV-Syn-EGFP was injected into the area adjacent to M2 at 6 weeks of age. Lentivirus for expression of Tau-shRNA, scrambled shRNA, B-Raf-shRNA, or Tyro3-siRNA, or plasmid vectors for expression of PKCα-shRNA, scrambled shRNA, or Gas6-shRNA dissolved in in vivo-jetPEI, were continuously injected into the subarachnoid space of the right M2 via osmotic pump from 8 to 12 weeks of age. Dendritic spines in layer 1 of M2 were observed by two-photon microscopy with three mice at 12 weeks of age. 8 to 10 images were obtained from one mice and the average of spine parameters were used for quantitative analyses ($N = 3$). The protocols for vemurafenib and Gö6976 were similar to those shown in Fig. 6 7, and two-photon microscopy was performed at 12 weeks of age. **b** Static spine morphology was observed by two-photon microscopy. Spine protrusion number, length, head diameter, and volume were analyzed. *$p < 0.05$; **$p < 0.01$ ($N = 3$, Student's t-test). Averages and s.e.m. are shown. **c** Images were obtained by two-photon microscopy at 0, 8, and 24 h on the last day of injection. Dynamic changes in spines were analyzed by serial observation. Spine formation and elimination were counted. *$p < 0.05$; **$p < 0.01$ ($N = 3$, Student's t-test). Averages and s.e.m. are shown. **d** Lentivirus for expression of shRNA-Tau was injected according to the protocol. **$p < 0.01$ ($N = 3$, Tukey's HSD test). Scrambled RNA was used as a control. Averages and s.e.m. are shown. **e** Lentivirus for expression of shRNA-B-Raf was injected according to the protocol. **$p < 0.01$ ($N = 3$, Tukey's HSD test). Scrambled RNA was used as a control. Averages and s.e.m. are shown. **f** Plasmid for expression of shRNA-PKCα was injected according to the protocol. *$p < 0.05$; **$p < 0.01$ ($N = 3$, Tukey's HSD test). Scrambled RNA was used as a control. Averages and s.e.m. are shown. **g** Lentivirus for expression of siRNA-Tyro3 was injected according to the protocol. Scrambled RNA was used as a control. **$p < 0.01$ ($N = 3$, Tukey's HSD test). Scrambled RNA was used as a control. Averages and s.e.m. are shown. **h** Plasmid for expression of shRNA-Gas6 was injected according to the protocol. Scrambled RNA was used as a control. **$p < 0.01$ ($N = 3$, Tukey's HSD test). Scrambled RNA was used as a control. Averages and s.e.m. are shown. **i** Mice received oral administration of vemurafenib (32 mg/kg of BW/day) from 6 to 12 weeks of age, and two-photon microscopic analysis was performed at 12 weeks. **$p < 0.01$ ($N = 3$, Tukey's HSD test). Averages and s.e.m. are shown. **j** Gö6976 (4.4 μM) or PBS was injected via osmotic pump into the subarachnoid space of PGRN-KI and C57BL/6J mice from 10 to 12 weeks of age, and two-photon microscopic analysis was performed at 12 weeks. **$p < 0.01$ ($N = 3$, Tukey's HSD test). Averages and s.e.m. are shown

at Thr220 was barely co-localized with PSD95 (Fig. 9a), and instead remained in the cytoplasm or dendrites (Fig. 1e), indicating that spine mislocalization of tau rarely occurs due to phosphorylation at Thr220. An increase in the abundance of

Ser191-phosphorylated tau in frontal cortex of PGRN-KI mice was detected using the AT8 antibody, but only at 48 weeks of age (Fig. 1e), which would be too late to trigger the synapse pathology observed at 12 weeks. The distinct consequences of different tau

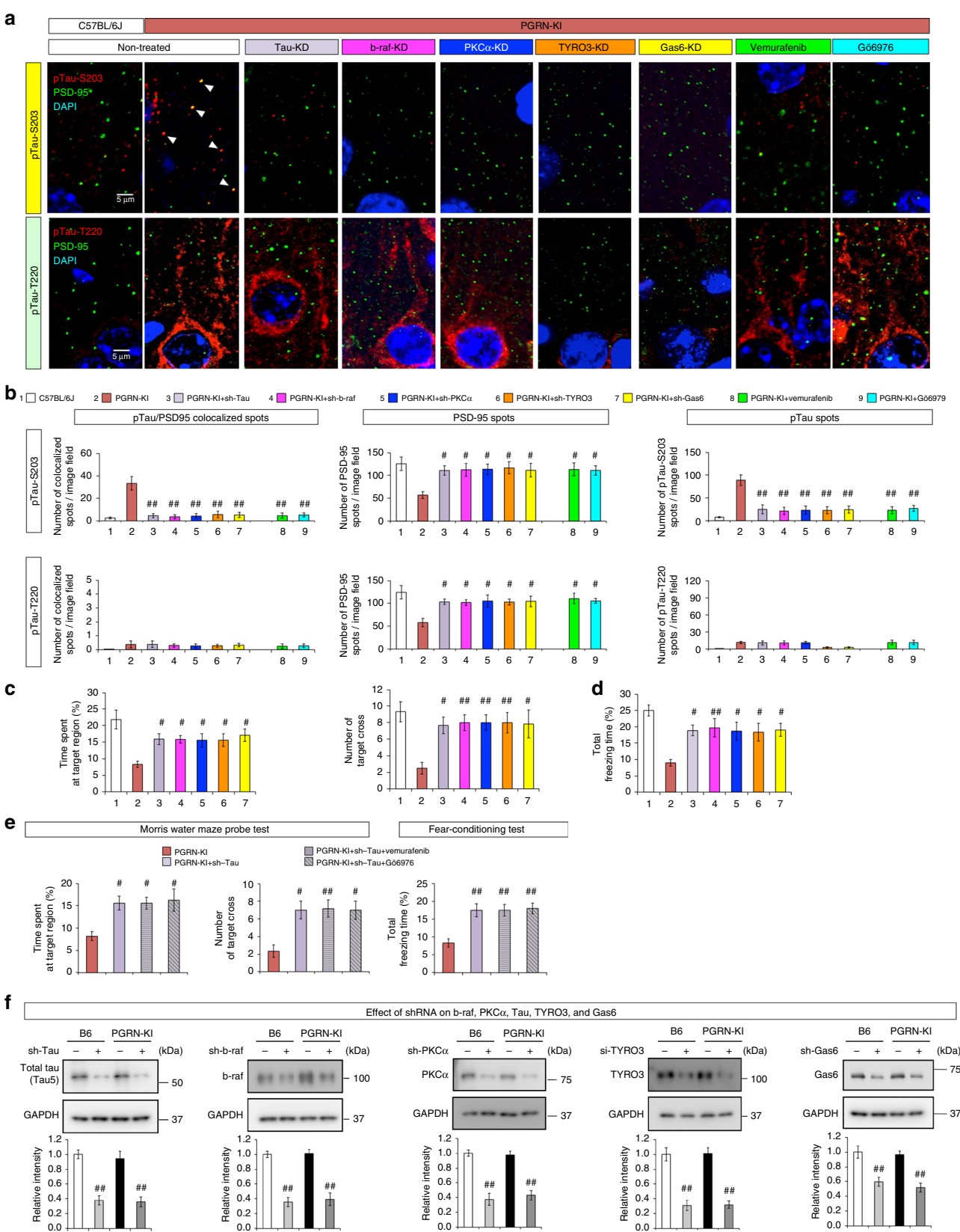

phosphorylation sites indicate that synapse pathology is specifically associated with phosphorylation at Ser203. These findings, together with the considerations described above, suggest that tau phosphorylation at Ser203 represents a promising target for delaying or stopping the early-stage pathology of FTLD before aggregation.

Based on the therapeutic effect of tau-KD on spine pathology (Figs. 8 and 9), tau protein could itself be a therapeutic target. However, tau protein has essential functions in neurons, and moreover a recent report suggested that the total amount of tau protein might be reduced in the brains of human patients with PGRN-linked FTLD[30]. Although reproducibility of these findings should be examined in a larger number of patients in the future, and our findings in mice suggested that a reduction of total tau is linked to normal aging rather than PGRN-FTLD pathology (Fig. 1f, g), we speculate that a reduction in tau levels might serve as a kind of protective response to maintain homeostasis under a pathological state. If so, exogenous suppression of total tau by KD would be unnecessary.

We showed that PKC and B-Raf inhibitors suppressed mislocalization of Ser203-phosphorylated tau to PSD95-positive spines, restored spine density, and rescued memory impairment in AD model mice. These findings are significant from the standpoint of the analogy to AD: a genome-wide association study suggested that the PKC pathway is activated in human AD patients[34], and a comprehensive phosphoproteome analysis of human AD and mouse AD model brains suggested that spine pathology is associated with PKC[31]. Accordingly, PKC and B-Raf represent drug seeds not only for FTLD, but also for AD. Before these targets are developed for clinical application, however, further investigation is necessary to determine how to selectively suppress the pathological, but not physiological, actions of B-Raf and PKC.

## Methods

**Ethics**. This study was performed in strict accordance with the Guidelines for Proper Conduct of Animal Experiments by the Science Council of Japan. It was approved by the Committees on Gene Recombination Experiments, Human Ethics, and Animal Experiments of the Tokyo Medical and Dental University (2010–215C3, 2011–22–3, and 0130225). For human studies, informed consent was obtained from all patients.

**Generation of PGRN R504X mutation-KI mice**. Knock-in mice harboring the heterozygous PGRN R504X mutation were constructed by insertion of a Neo cassette provided by Unitech (Chiba, Japan) into C57BL/6J mice. The targeting vector was generated from two constructs. (Construct 1) A PCR product of a 6.5 kbp NotI–XhoI fragment containing the R504X mutation, amplified from a BAC clone (ID: RP23-311P1 or RP23-137J17), was subcloned into vector pBS-DTA

(Unitech). (Construct 2) A PCR product of a 3.0 kbp ClaI–XhoI fragment was subcloned into vector pBS-LNL(−) containing the Neo cassette. The BamHI (Blunt)–XhoI fragment from Construct 2 was subcloned between the XhoI (Blunt) and SalI sites of Construct 1, and the resultant plasmid was used as the targeting vector. After linearization with SwaI, the fragment was electroporated into a C57BL/6J mouse ES clone. Genotyping of ES clones was performed by PCR using the following primers: 5′-CGTGCAATCCATCTTGTTCAAT-3′ (forward), 5′-CATGACCTAACTCAATGCATACCAC-3′ (reverse). Positive clones were subjected to Southern blot analysis. The 5′ and 3′ probes for Southern blots were generated as follows: 5′-probe, 5′-CTGTGTCTCACTAGAAGCATAAGCA-3′ (forward), 5′-ACTAGATTGGGAAAGACAGTGAATC-3′ (reverse); 3′-probe, 5′-AATTCCAATCCTGTGTGGTCATAG-3′ (forward), 5′-CCACTTTCTTTCT CCCTTACCCTA-3′ (reverse). Neomycin probes were generated using the following primers: 5′-GAACAAGATGGATTGCACGCAGGTTCTCCG-3′ (forward), 5′-GTAGCCAACGCTATGTCCTGATAG-3′ (reverse). The Neo cassette, present in the F1 mice, was removed by crossing with CAC-Cre mice.

The presence of the Pgrn mutation in the knock-in mice was verified by PCR using genomic DNA prepared from tail snips. Amplification was performed using KOD-FX neo (TOYOBO, Japan) using the following primers: 5′-GATCTTTCT ATGTACAGCCACGTTT-3′ (forward), 5′-CTAATGGTGTTCAACGTCAAG TAGT-3′ (reverse). PCR conditions were as follows: 35 cycles of 98 °C for 10 s (denaturation), 57 °C for 30 sec (annealing), and 68 °C for 45 s (extension). Only a 302 bp fragment was amplified from the genome of background mice, whereas the 302 bp wild-type fragment and a 392 bp fragment containing LoxP were amplified from heterozygous KI mice.

Genotyping was performed by multiple examiners, without exchange of information, prior to behavioral tests. The results from each examiner were mixed for statistical tests, with power analysis to validate sample size. All animal experiments were performed in accordance with the 'Animal Research: Reporting of In Vivo Experiments' (ARRIVE) guidelines[74].

**Behavioral analyses**. Six types of behavioral tests were performed for male mice, as described previously[75], at the ages indicated in figures or legends. In the Morris water maze test, mice performed four trials (60 s) per day for 5 days, and latency to reach the platform was measured. Also, on day 5, a probe test was performed in which the mouse's movement without the platform was tracked for 60 s, and the time spent in the platform region was measured. In the rotarod test, mice performed four trials (3.5 to 35 r.p.m.) per day for 3 days, and the mean latency of falling off the rotarod was recorded. In the fear-conditioning test, the freezing response was measured at 24 h after the conditioning trial (65 dB white noise, 30 s + foot shock, 0.4 mA, 2 s) in the same chamber in the absence of a foot shock. In the open-field test, the duration of the stay in the central area of the open-field (50 × 50 × 40 cm [H]) was adopted as the index. In the light–dark box test, the duration of the stay in the light box (20 × 20 × 20 cm [H]) was measured. The elevated plus maze was set up 60 cm above the floor, and the duration of the stay in the open arms was measured. Animal behavioral experiments were performed by multiple examiners.

**Western blot analysis**. Mouse cerebral cortex tissues were washed three times with ice-cold PBS and dissolved in lysis buffer containing 62.5 mM Tris-HCl, pH 6.8, 2% (w/v) SDS, 2.5% (v/v) 2-mercaptoethanol, 5% (v/v) glycerol, and 0.0025% (w/v) bromophenol blue. Samples from cultured cells were prepared similarly. Protein concentration was quantified using the Pierce BCA Protein Assay Kit (Thermo Scientific, Waltham, MA, USA). Samples were separated by SDS-PAGE, transferred onto Immobilon-P polyvinylidene difluoride membrane (Millipore, Billerica, MA, USA) through a semi-dry method, and blocked with 5% milk in

**Fig. 9** Suppression of Tyro3 signal rescues tau mislocalization and cognitive impairment in mutant PGRN-KI mice. **a** Co-staining of phosphorylated tau (Ser203 or Thr220) and PSD-95. Confocal microscopic analysis of M2 regions of background (B6, C57BL/6J), PGRN-R504X-KI (PGRN-KI), vemurafenib-treated PGRN-KI, Gö6976-treated PGRN-KI, and lentivirus-tau-shRNA–infected PGRN-KI mice. All images were acquired by confocal microscopy (LSM510META, Carl Zeiss, Germany). Z-stack images were acquired with the following parameters; objective: ×63, average: line 2; filter (Cy3 ChS1: 550–679 nm; FITC Ch1: 505–530 nm; DAPI Ch2: 420–480 nm); master gain (Cy3 ChS1: 760; FITC Ch1: 741; DAPI Ch2: 615). **b** Quantitative analyses of PSD95-positive dots and p-tau/PSD95 double-positive spots. All bar graphs indicate averages and s.e.m. Three mice were used for each group. The number of spots was counted in 10 image fields (100 × 100 μm) for each mouse, and the average was used for calculation of mean with s.e.m. for each group. The number of p-tau/PSD95 double-positive spots increased in PGRN-KI mice. #$p < 0.05$, ##$p < 0.01$ ($N = 6$, Dunnett's test). Averages and s.e.m. are shown. **c** In vivo effects of KD of B-Raf, PKCα, Tyro3, Gas6, and tau on two parameters (% time spent at target region and number of target crosses) in the Morris water maze test, performed with PGRN-KI mice ($N = 6$ in each experiment) at 12 weeks. #$p < 0.05$, ##$p < 0.01$ ($N = 6$, Dunnett's test). Averages and s.e.m. are shown. **d** In vivo effect of KD of B-Raf, PKCα, Tyro3, Gas6, and tau on % total freezing time in the fear-conditioning test, performed with PGRN-KI mice ($N = 6$ in each experiment) at 12 weeks. #$p < 0.05$, ##$p < 0.01$ ($N = 6$, Dunnett's test). Averages and s.e.m. are shown. **e** Tau-KD with or without vemurafenib/Gö6976 had similar effects on % time spent at target region and number of target crosses in the Morris water maze test and on % total freezing time in fear-conditioning test. Both tests were performed at 12 weeks. *$p < 0.05$ ($N = 6$, Dunnett's test). Averages and s.e.m. are shown. **f** Western blot analysis confirming shRNA-mediated knockdown of B-Raf, PKCα, and tau. Lower graphs show quantitation. P-values were determined by Tukey's HSD test ($N = 6$). Averages and s.e.m. are shown. p-values are shown in Supplementary data 1

## a

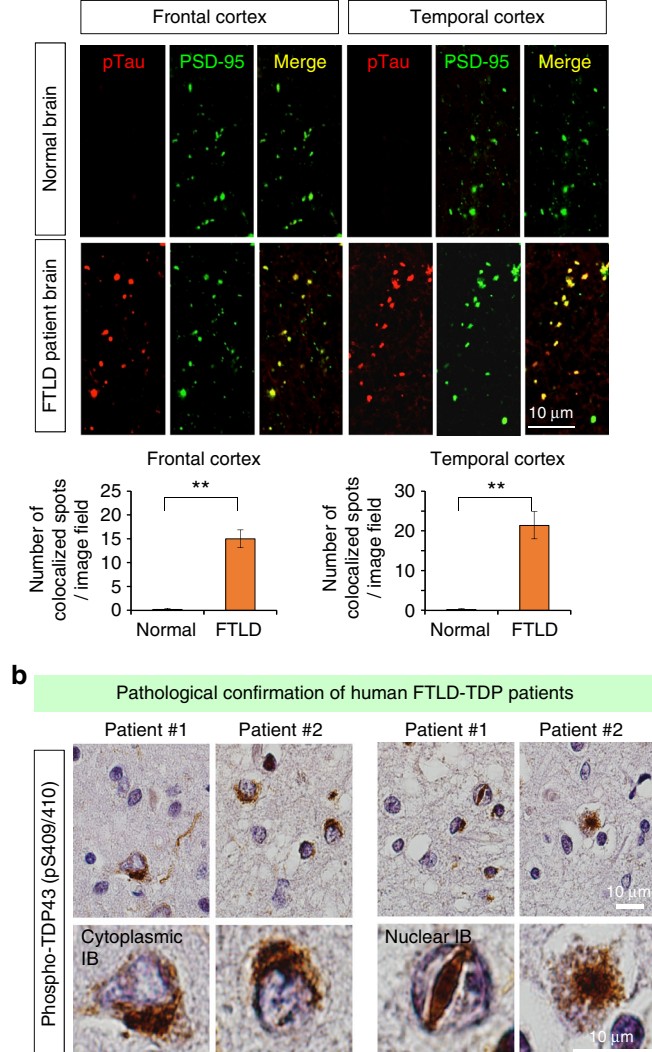

pSer214-Tau (=mouse pSer203) in sporadic human FTLD-TDP

Fig. 10 Mislocalization of human pSer214 to spine. **a** Immunohistochemistry of pSer214-tau, which corresponds to mouse pSer203-tau, in human sporadic FTLD-TDP autopsy samples revealed similar mislocalization of specific types of tau to PSD95-positive spines (left images). Averages and s.e.m. are shown. Five independent cases were examined, and the averaged values of 10 visual fields (40 × 40 μm) in each case were used for quantitative analysis (right graphs). *P*-values were determined by Student's *t*-test (*N* = 5). **p < 0.01. **b** Pathological diagnosis of patients was based on the presence of TDP43. Findings from each case are shown. *p*-values are shown in Supplementary data 1

TBST (10 mM Tris/Cl, pH 8.0, 150 mM NaCl, 0.05% Tween20). Primary and secondary antibodies were diluted in TBST with 0.5% skim milk or Can Get Signal solution (Toyobo) as follows: anti-PGRN, 1:1000 (AF2557, R&D Systems, Minneapolis, MN, USA); anti-PGRN, 1:200 (ST1574, Calbiochem); anti-phosphorylated-B-Raf (human Ser729 and mouse Ser766), 1:9000 (ab124794, Abcam, Cambridge, UK); anti-phosphorylated-PKCγ (human/mouse Thr655), 1:5000 (ab5796, Abcam); anti-phosphorylated-MEK-1 (human/mouse Thr386), 1:10000 (07–853, Millipore); anti-MEK1, 1:3000 (ab32088, Abcam); anti-phosphorylated-ERK1/2, 1:5000 (#4370, Cell Signaling); anti-ERK1/2, 1:5000 (#9102, Cell Signaling); anti-phosphorylated-PKCα, 1:3000 (Ser319, AP0560, NeoScientific, Cambridge, MA, USA); anti-TRADD (sc-7868, Santa Cruz Biotechnology, Dallas, TX, USA), 1:400; anti-RIP1 (MAB3585, R&D), 1:500; anti-TRAF2 (sc-877, Santa Cruz Biotechnology), 1:2000; anti-caspase-3 (9662, Cell Signaling Technology, Danvers, MA, USA), 1:1000; anti-GAPDH, 1:1000–10,000 (MAB374, Millipore); HRP-linked anti-rabbit IgG, 1:3000 (NA934, GE Healthcare); HRP-linked anti-mouse IgG, 1:3000 (NA931, GE Healthcare). Blots were incubated with primary

and secondary antibodies overnight at 4 °C and for 1 h at room temperature, respectively. ECL Prime Western Blotting Detection Reagent (RPN2232, GE Healthcare) and a luminescence image analyzer (ImageQuant LAS 500, GE Healthcare) were used to detect proteins. Uncropped western blots are shown in Supplementary Fig. 8.

**Sarkosyl-insoluble fraction.** Detection of phosphorylated TDP43 in the sarkosyl-insoluble fraction was performed as previously described[17,62]. Frozen whole cerebral cortexes (50 mg) of PGRN-KI and littermate (C57BL/6) mice were homogenized in a 7 ml Dounce tissue grinder on ice with 5 ml of extract buffer A (10 mM Tris-HCl (pH 7.5), 1 mM EGTA, 10% sucrose, 0.8 M NaCl). After addition of another 5 ml of extract buffer A containing 2% Triton X-100 and incubation for 30 min at 37 °C, the homogenates were centrifuged at 100,000 × g for 30 min at 4 °C. Pellets were dissolved in 200 μl of buffer A containing 1% sarkosyl, incubated for 30 min at room temperature, and centrifuged at 100,000 × g for 30 min. The pellet was dissolved with 500 μl of buffer A containing 1% CHAPS, incubated for 30 min at 37 °C, and centrifuged at 100,000 × g for 20 min at room temperature. The resultant pellets were dissolved and sonicated with 100 μl of 7 M guanidine hydrochloride and dialyzed against 30 mM Tris-HCl (pH 7.5) overnight. No kinase or phosphatase inhibitors were used during incubation or dialysis, as recommended in previously published protocols[17,62], and therefore it cannot be excluded that TDP43 phosphorylation or de-phosphorylation changed during the experimental manipulations. The dialyzed samples were mixed with equal volumes of sample buffer (62.5 mM Tris-HCl, pH 6.8, 2% (w/v) SDS, 2.5% (v/v) 2-mercaptoethanol, 5% (v/v) glycerol, and 0.0025% (w/v) bromophenol blue).

**Preparation of phosphorylated proteins and peptides.** Cerebral cortices were collected from mice within 5 min after sacrifice by deep euthanasia with ethyl ether. Cerebral cortices were immediately frozen in liquid nitrogen and kept under the same condition until use. Cortical tissue was lysed in cold lysis buffer containing 2% SDS, 1 mM DTT, and 100 mM Tris-HCl (pH 7.5), and then homogenized by 20 strokes of a Dounce glass homogenizer on ice. The ratio of lysis buffer to tissue was 10 volumes (μl) to 1 weight (mg). Subsequently, the lysate was incubated at 100 °C for 15 min. The crude extract was centrifuged at 16,000 g at 4 °C for 10 min, and the resultant supernatant was diluted 1:10 with water and filtered through a 0.22 μm filter. The flow-through was concentrated 10-fold using an Amicon Ultra 3 K filter (Millipore, Ireland). Protein concentration was measured using the Pierce BCA Protein Assay Reagent. Sample aliquots (200 μl) containing 1.5 mg of protein were mixed with 100 μl of 1 M triethylammonium bicarbonate (TEAB) (pH 8.5), 3 μl of 10% SDS, and 30 μl of 50 mM tris-2-carboxyethyl phosphine (TCEP), and then incubated for 1 h at 60 °C. Cysteine residues were blocked with 10 mM methyl methanethiosulfonate (MMTS) for 10 min at 25 °C. The samples were then digested with trypsin (mass analysis grade) (10:1 protein/enzyme, w/w) in 80 mM CaCl₂ for 24 h at 37 °C.

Phosphopeptides were enriched using the Titansphere Phos-Tio Kit (GL Sciences, Tokyo, Japan), and desalted using a Sep-Pak Light C18 cartridge column (Waters Corporation, Beverly, MA, USA). The sample aliquots were dried and dissolved with 25 μl of 100 mM TEAB (pH 8.5). The phosphopeptides in each sample were labeled separately for 2 h at 25 °C using the iTRAQ Reagent multiplex assay kit (AB SCIEX). The labeled phosphopeptide pools were then mixed together. The aliquots obtained were dried and redissolved in 1 ml of 0.1% formic acid.

**2D LC MS/MS analysis.** The labeled phosphopeptide samples were subjected to Strong Cation Exchange (SCX) chromatography using a TSK gel SP-5PW column (TOSOH, Japan) on a Prominence UFLC system (Shimadzu, Japan). The flow rate was 1.0 ml/min with solution A (10 mM KH₂PO₄ (pH 3.0), 25% acetonitrile). Elution was performed with solution B (10 mM KH₂PO₄ (pH 3.0), 25% acetonitrile, 1 M KCl) in a gradient ranging from 0 to 50%. The elution fractions were dried and dissolved in 100 μl of 0.1% formic acid.

Each fraction was analyzed using a DiNa Nano-Flow LC system (KYA Technologies Corporation, Japan) at a flow rate of 300 nl/min. For Nano-LC, samples were loaded onto a 0.1 × 100 mm C18 column with solution C (2% acetonitrile and 0.1% formic acid) and eluted with a gradient of 0–50% solution D (80% acetonitrile and 0.1% formic acid). The ion spray voltage for application of sample from the Nano-LC to the Triple TOF 5600 System (AB SCIEX Ins.) was set at 2.3 kV. The Information-Dependent Acquisition (IDA) setting was 400–1250 m/z with two to five charges. Analyst TF software (version 1.5, AB SCIEX) was used to identify each peptide. Quantification of each peptide was based on the TOF-MS electric current detected during the LC-separated peptide peak, adjusted to the charge/peptide ratio.

**Data analysis.** Mass spectrum data of peptides were acquired and analyzed using Analyst TF (version 1.5, AB SCIEX). Based on the results, corresponding proteins were retrieved from public databases of mouse and human protein sequences (UniProtKB/Swiss-Prot, downloaded from http://www.uniprot.org on 22 June 2010) using ProteinPilot (version 4, AB SCIEX), which employs the Paragon algorithm[76]. In ProteinPilot, tolerance of peptide search was set to 0.05 Da for MS and 0.10 Da for MS/MS analyses. 'Phosphorylation emphasis' was set at sample description, and 'biological modifications' was set at the processing specification

step. The confidence score was used to evaluate the quality of identified peptides, and the deduced proteins were grouped using the Pro Group algorithm (AB SCIEX) to eliminate redundancy. The threshold for protein detection in ProteinPilot was set at 95% confidence, and proteins with >95% confidence were accepted as identified proteins.

Quantification of peptides was performed by analysis of iTRAQ reporter groups in MS/MS spectra generated upon fragmentation in the mass spectrometer. For bias correction, signals of different iTRAQ reporters were normalized based on the assumption that the total signal of all iTRAQs should be equal. After bias correction, the ratio between reporter signals in PGRN-KI and control mice (peptide ratio) was calculated.

The protein ratio was computed from the weighted average of peptide ratios corresponding to each protein; the peptide ratios were differentially weighted based on error factors after bias correction. The detailed formulas for calculation of these values are described in the AB SCIEX manual. In brief, after excluding peptides without iTRAQ label, those sharing MS/MS derived from other proteins, and those with low intensity, log values of iTRAQ ratios corresponding to a peptide were summed, and bias was divided by 10 raised to the power of the sum. The result was treated as the quantity of the peptide.

For further data analyses, peptide and protein ratios were imported into Excel files from the ProteinPilot summaries. The quantity of a phosphopeptide fragment was calculated as the geometric mean of signal intensities of multiple MS/MS fragments including the phosphorylation site. Because the ratios of phosphoproteins and phosphopeptides obeyed a log-normal distribution, the ratios were transformed into logarithmic values to compare the PGRN-KI and control groups. Because the data were postulated to be heteroscedastic, the difference between PGRN-KI and controls was statistically evaluated by Welch's test.

**Generation of PPI networks**. To create lists for generation of PPI networks, changes in the levels of phosphopeptides at each time point were compared between PGRN-KI and control mice by Welch's test with post-hoc BH procedure. Significantly changed phosphopeptides ($q < 0.05$) were used to select the corresponding proteins. UniProt IDs were added to the proteins altered in phsophorylation. Proteins whose UniProt IDs were not listed in the Human Genome Project (GNP) (http://genomenetwork.nig.ac.jp/index_e.html) database were removed from the list of altered proteins. The selected proteins were used for generation of the pathological PPI network of PGRN-KI at each time point, based on the integrated database of GNP including BIND (http://www.bind.ca/), BioGrid (http://www.thebiogrid.org/), HPRD (http://www.hprd.org/), IntAct (http://www.ebi.ac.uk/intact/site/index.jsf), and MINT (http://mint.bio.uniroma2.it/mint/Welcome.do). One additional edge and node were added to the selected nodes on the PPI database. A database of GNP-collected information was created on the Supercomputer System available at the Human Genome Center of the University of Tokyo. The resultant PPI network was visualized using Cell Illustrator[77], and representative quantity (geometric mean of peptide ratios of significantly changed phosphopeptides) and lowest q-value were attached to each node.

**Identification of altered signaling pathways**. Lists of genes involved in various signal pathways were acquired from the KEGG Pathway database (http://www.genome.jp/kegg/pathway.html). Phosphorylated proteins for which phosphorylation of specific sites changed significantly in whole cerebral cortex of PGRN-KI mice ($q < 0.05$, Welch's test with post-hoc BH procedure) were mapped onto the selected pathways using the REST-style KEGG API (http://www.kegg.jp/kegg/rest/keggapi.html), which provides ID mapping functions that enable acquisition of UniProt IDs from KEGG gene identifiers. Fisher's exact test was performed to evaluate whether a given pathway was selectively altered in PGRN-KI mice by the ratio of changed phosphoproteins between whole proteins and pathway-specific proteins.

**Effect of vemurafenib on B-Raf downstream pathways**. The B-Raf downstream pathway was constructed using a PPI database compiled by the Genome Network project (http://genomenetwork.nig.ac.jp/index_e.html). Factors one or two degrees of separation from B-Raf (UniProt ID = P28028) in the database were selected as candidate nodes and edges in the B-Raf downstream pathway. Phosphorylation sites suppressed by vemurafenib were determined from phosphoproteome data by comparison between vemurafenib-treated and non-treated PGRN-KI mice.

Whole cortex tissues from three vemurafenib-treated PGRN-KI mice and three non-treated PGRN-KI mice, prepared at 4 weeks of age, were analyzed by mass spectrometry (Nano-LC to Triple TOF 5600 System, AB SCIEX); specimens from pooled B6 mice of the same age were used as controls. Phosphopeptides detected at >95% confidence were quantified relative to the control as described above, and values were integrated on specific phosphorylation sites. Proteins with detected phosphorylation site(s) were defined as detected phosphoproteins. Suppressed phosphorylation sites were defined by the following criteria: (1) significantly decreased in abundance between two groups ($q < 0.05$, Welch's test with post-hoc BH correction), or (2) quantified in non-treated PGRN-KI mice but not detected in vemurafenib-treated PGRN-KI mice. The phosphoprotein value was generated by the weighted average of phosphorylation site values corresponding to each protein. The specificity of vemurafenib's effect on B-Raf downstream proteins was tested by

Fisher's exact test of suppression ratio in comparison with B-Raf-non-related proteins.

**Two-photon microscopic analysis of spine pathology**. This procedure was described in detail elsewhere[78]. Adeno-associated virus 1 (AAV1)-EGFP with the synapsin I promoter (titer, $1 \times 10^{10}$ vector genomes/ml; 1 µl) was injected into the retrosplenial cortex (anteroposterior, −2.0 mm from Bregma; lateral 0.6 mm; depth, 1 mm) of 10-week-old mice under anesthesia with 2.5% isoflurane. Two weeks later, a high-speed micro-drill was used to thin a circular area of skull, whose center was at anteroposterior, −0.6 mm from Bregma and 0.5 mm lateral from midline that corresponds to the surface of M2 cortex. Then, the head of the animal was immobilized by attaching the head plate to a custom-machined stage mounted on the microscope table. Two-photon imaging was performed using a laser-scanning microscope system FV1000MPE2 (Olympus, Japan) equipped with an upright microscope (BX61WI, Olympus, Japan), a water-immersion objective lens (XLPlanN25xW; numerical aperture, 1.05), and a pulsed laser (MaiTaiHP DeepSee, Spectra Physics, USA). EGFP was excited at 890 nm and scanned at 500−550 nm. High-magnification imaging (101.28 × 101.28 µm; 1024 × 1024 pixels; 1 µm Z-step) of cortical layer I was performed at 5 × digital zoom through the window in retrosplenial cortex.

**Rescue of spine morphology and behavioral analyses**. In pharmacological rescue experiments, mice were fed vemurafenib (32 mg/kg of BW/day) or mock from 6 to 12 weeks of age, or received 4.4 µM Gö6976 (Calbiochem, San Diego, CA, USA) or PBS into the subarachnoid space of right M2 via osmotic pump (0.15 µl/h, #2006, DURECT, Palo Alto, CA, USA) from 10 to 12 weeks of age. Images were obtained by two-photon microscopy at 0, 8, and 24 h on the last day of injection.

In KD rescue experiments, lentiviral particles ($2 \times 10^5$ titer units in 100 µl) for Tau-shRNA (sc-430402-V, Santa Cruz Biotechnology), scrambled shRNA (SC-108080, Santa Cruz Biotechnology), B-Raf shRNA (sc-63294-V, Santa Cruz Biotechnology), or Tyro3-siRNA (iV037848, ABM, Richmond, BC, Canada) diluted 1:4 with ACSF buffer (NaCl, 125 mM; KCl, 2.5 mM; NaH$_2$PO$_4$, 1.25 mM; MgCl$_2$, 1 mM; CaCl$_2$, 1 mM; NaHCO$_3$, 26 mM; glucose, 25 mM) or plasmid vectors (10 µg) for PKCα-shRNA (TG501653, OriGene, Rockville, MD, USA), scrambled shRNA (TG501653, OriGene), or Gas6-shRNA (sc-35451-SH, Santa Cruz Biotechnology) dissolved in 100 µl of in vivo-jetPEI (201−10 G, Polyplus-transfection, Illkirch, France) were injected into the subarachnoid space of the right RSD via osmotic pump (0.15 µl/h) from 8 to 12 weeks of age.

For spine morphology analysis in 12-week-old mice, dendrite images were obtained by two-photon microscopy in the first layer of the M2, 2−3 mm away from the injection point in *XY* plane, and analyzed for spine density, length, head diameter, and volume using the IMARIS software (IMARIS7.2.2, Bitplane, Switzerland).

In behavioral analyses of 12-week-old mice, two parameters (% time spent at target region and number of target crosses in the Morris water maze test and % total freezing time) were used to evaluate rescue effects.

**Immunohistochemistry**. For immunohistochemistry, brain samples were fixed with 4% paraformaldehyde and embedded in paraffin. Sagittal or coronal sections (5 µm thickness) were obtained using a microtome (Yamato Kohki Industrial, Tokyo, Japan). Immunohistochemistry was performed using the following primary antibodies: anti-PGRN antibody, AF2557, R&D Systems, 1:200; anti-NeuN antibody, MAB377B, Millipore, 1:100; anti-TDP43 antibody, G400, Cell Signaling Technology, 1:500; anti-ubiquitin antibody, Cell Signaling Technology, P4D1, 1:1000; anti-phospho-TDP43 antibody, #TIP-PTD-P02, Cosmo Bio, 1:500; anti-FUS antibody, ab84078, Abcam, 1:200; anti-p62 antibody, #610497, BD Bioscience, 1:200; anti-IBA1 antibody, 019−19741, WAKO, 1:500; Cy3-conjugated anti-GFAP antibody, C9205, Sigma-Aldrich, 1:1000; anti-CD4 antibody, sc-13573, Santa Cruz Biotechnology, 1:200; anti-B2M antibody, 13511−1-AP, Proteintech, 1:200; anti-CD8 antibody, sc-7188, Santa Cruz Biotechnology, 1:200; anti-γH2AX antibody, JBW301, Millipore, 1:100; anti-PHF-Tau (AT-8), Innogenetics, 1:500; anti-phospho-tau T231 antibody, ab151559, Abcam, 1:200: anti-phospho Tau S214, ab170892, Abcam, 1:200; anti-PSD-95 antibody, MA1−045, Pierce, 1:200. Reaction products were visualized with Alexa Fluor 488− or Alexa Fluor 568−conjugated secondary antibodies. TDP43 and ubiquitin were visualized with the Vectastain Elite ABC kit and DAB Peroxidase Substrate kit (Vector, Burlingame, CA, USA). Nuclei were stained with DAPI (DOJINDO, 0.2 µg/ml in PBS). Images were acquired on an Olympus IX70 fluorescence microscope (Tokyo, Japan) or an LSM510 Meta confocal microscope (Zeiss).

Antibody absorption test was performed with the following human tau peptides: Ser214 (mouse Ser203)-phosphorylated peptide (ab191123, Abcam), Thr231 (mouse Thr220)-phosphorylated peptide (#51110, Signalway, College Park, MD, USA), and Thr205 (mouse Thr194)-phosphorylated peptide (ab5249, Abcam). Peptide and antibody were mixed at a molar ratio of 30:1 and incubated for 12 h at 4 °C. After centrifugation (12,000 rpm for 10 min at 4 °C), the supernatants were collected and applied to sections.

**Mislocalization of tau to spines**. Immunostaining with anti-PSD-95 and phosphorylated tau (Ser203 or Thr220) was performed with paraffin sections (5 μm) from M2 tissues and observed on an LSM510 Meta confocal microscope (objective, ×63; zoom 1; 10 z-stack images at 0.8 μm intervals). For co-localization analysis, merged dot signals of phospho-tau and PSD-95 in acquired images (40 × 40 μm) were counted. In addition, fluorescence signal intensities of phosphorylated tau or PSD-95 were quantified using ZEN Lite 2012 (Zeiss), and mean pixel intensities in each ROI (20 × 20 μm) were calculated. Data were acquired from 10 randomly selected images, and comparisons among mouse groups were performed by one-way ANOVA with Tukey's test for multiple comparisons.

**Surface plasmon resonance**. SPR analysis was performed on a Biacore T100 instrument (GE Healthcare, USA). For binding of Tyro3 and Gas6, 25 μg/ml anti-Fc antibody (BR-1008–39, GE Healthcare) was immobilized to 10,000 resonance units onto a CM5 sensor chip in HBS-EP with 1.5 mM CaCl$_2$ according to the standard protocol (GE Healthcare). Next, 50 μl of 14 μg/μl Fc-Tyro3 (D0315, SIGMA, MI, USA) was injected over 25 min at a rate of 2 μl/min. Then, Gas6 (885-GSB-050, R&D Systems) at the indicated concentration was injected at the same flow rate for 25 min. Before co-injection of PGRN and Gas6, the indicated concentrations of full-length PGRN (2420-PG-050, R&D Systems) and Gas6 were incubated at 4 °C or 37 °C for 6 h. The conditions for PGRN/Gas6 injection were the same as those for Gas6 injection. For regeneration of the sensor chip, 3 M MgCl$_2$ (GE Healthcare) was injected at a rate of 10 μl/min for 1 min.

For binding of PGRN and Gas6, 30 μg/ml anti-Gas6 antibody (AF885-SP, R&D, MN, USA) was immobilized to 6600 resonance units. Next, 500 μM Gas6 was injected at a rate of 10 μl/min for 5 min. Then, the indicated concentrations of PGRN were injected at the same flow rate. For regeneration of the sensor chip, 10 mM glycine-HCl (pH 1.7) was injected at the same flow rate for 1 min.

**Pull-down assay**. His-PGRN (CF2420-PG-050, R&D Systems) or His-BDNF (MBS962271, MyBioSource, CA, USA), and GST-Gas6 (H00002621-P01, Abnova, Taipei City, Taiwan) or GST prepared as described previously[79], were mixed in 200 μl of HBS-EP containing 1.5 mM CaCl$_2$ and incubated at 4 °C for 6 h. After addition of 50 μl of Ni-NTA agarose (30210, Qiagen, Hilden, Germany) or glutathione–Sepharose (17075601, GE Healthcare), the mixtures were incubated for 3 h at 4 °C, centrifuged, and washed five times with PBS. The beads were mixed with an equal volume of sample buffer (62.5 mM Tris-HCl, pH 6.8, 2% (w/v) SDS, 2.5% (v/v) 2-mercaptoethanol, 5% (v/v) glycerol, and 0.0025% (w/v) bromophenol blue) and boiled at 95 °C for 10 min.

**Identification of kinases responsible for tau phosphorylation**. Human tau cDNA was kindly provided by Drs Y. Kanai and N. Hirokawa (University of Tokyo). For in vitro analysis, GST-Tau (0.5 μM) was incubated for 20 min at 30 °C with each kinase (30 nM) in a reaction mixture consisting of 25 mM Tris-HCl, pH 7.5, 1 mM EDTA, 1 mM DTT, 5 mM MgCl$_2$, and 50 μM γ-$^{32}$P-ATP (15 GBq/mmol). The reaction mixtures were boiled in SDS sample buffer and subjected to SDS-PAGE. Radiolabeled proteins were analyzed on an FLA9000image analyzer (GE Healthcare, Little Chalfont, UK).

To assess phosphorylation in vivo (i.e., in cells), plasmids encoding tau and each kinase in the following were transfected into COS7 cells (purchased from ATCC, CRL-1651) using the Lipofectamine 2000 reagent (Thermo Fisher Scientific). Absence of mycoplasma infection was confirmed routinely. The cells were lysed with SDS sample buffer and subjected to immunoblot analyses. The following plasmids were used: pEGFP-c1-Tau (CMV promoter, EGFP-human Tau), pEF-BOS-GST-MEK1-CA (EF-1α promoter, GST-human MEK1-CA), pEF-BOS-GST-ERK2 (EF-1α promoter, GST-human ERK2), pEF-BOS-GST-PKA-CA (EF-1α promoter, GST-mouse PKA-CA), pEF-BOS-GST-Rho-kinase-cat (EF-1α promoter, GST-bovine Rho-kinase catalytic region), pEF-BOS-GST-CDK5 (EF-1α promoter, GST-human CDK5), pEF-BOS-GST-p35 (EF-1α promoter, GST-human p35), pCGN-HA-GSK3B-CA (CMV promoter, HA-human GSK3B-CA), pEF-BOS-GST-CaMK1-cat (EF-1α promoter, GST-rat CaMK1 catalytic region), pEF-BOS-GST-CaMK2-cat (EF-1α promoter, GST-rat CaMK2 catalytic region), pEF-BOS-GST-PKCα-cat (EF-1α promoter, GST-rat PKCα catalytic region), pEF-BOS-GST-MARK1-cat (EF-1α promoter, GST-human MARK1 catalytic region), pEF-BOS-GST-DCLK1-cat (EF-1α promoter, GST-human DCLK1 catalytic region), and pEF-BOS-HA-PKN-cat (EF-1α promoter, GST-human PKN1 catalytic region).

**Randomization of samples and animal selection**. Selection of animals and the behavior/weight analyses were performed by independent researchers. Randomization (selection) of animals was determined based on chronological order of birth date. Selection of images from immunohistochemistry and actual IHC experiments was performed by different researchers. Investigators were blinded to the group allocation during the experiment.

**Data availability**. The source data depicted in Supplementary Figs. 2B, 3A, 3C and 5 are available in an online database (http://suppl.atgc.info/011/), and the rest of the data are available within the article and Supplementary Information or available from the authors upon reasonable request.

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

## Acknowledgements

This work was supported by Brain Mapping by Integrated Neurotechnologies for Disease Studies (Brain/MINDS) from the Japan Agency for Medical Research and Development (AMED), the Strategic Research Program for Brain Sciences (SRPBS), and a Grant-in-Aid for Scientific Research on Innovative Areas 'Foundation of Synapse and Neurocircuit Pathology' (22110001, 22110002) from the Ministry of Education, Culture, Sports, Science and Technology (MEXT) and CREST from the Japan Science and Technology Agency (JST) (to H.O.). We thank our lab members, Drs Takuya Tamura, Chisato Yoshida, Tsutomu Oka, Hikaru Ito, Toshikazu Sasabe, Akiko Ohtani, Yuji Ogushi, and

Kazumi Motoki, and Ms. Tayoko Tajima (Neuropathology, TMDU) for technical support.

## Author Contributions

K.F., X.C., H.H., K.T. and M.A. designed and performed experiments, analyzed the data, and wrote the paper. A.S., S.I., K.K. and S.M. analyzed data. H.A. and Y.H. prepared human materials. H.O. planned the project, co-designed the experiments, and wrote the paper.

## Additional information

**Competing interests:** The authors declare no competing financial interests.

