## [Peer Review File · Nature Communications]

Reviewers' comments:

Reviewer #1 (Remarks to the Author):

The manuscript by Fujita et al uses a variety of methods to uncover a potential new pathway in PGRN-mediated FTLD. First, they developed a knockin mouse model for progranulin, which surprisingly shows TDP-43 pathology. Second, they use phospho-proteomics to analyze altered signaling in these mice. Third, this leads to discovery of TYRO3 a putative novel PGRN-receptor, or more precisely a ligand/receptor pair inhibited by PGRN. Fourth, they use two candidate drugs to prevent altered signaling in the mouse model, which apparently improves cognition. One of the drugs, Vemurafenib, is FDA approved for cancer with the V600E b-raf mutation, because it is thought to be specifically acting on the mutant protein. There it is unclear if the drug is safe at a dose necessary for efficacy in mice/patients with wild-type b-raf. Cramming this much data in one paper is impressive, but also prevents a thorough analysis of many crucial points and thus hurts the manuscript overall.

-The manuscript is readable but would greatly benefit from English language editing. There are many typos (e.g. "PGN" and "moths") and weird sentences. Some words are used in the wrong context and far too frequently (e.g. "obviously", "actually").

-Statistical analysis is weak and poorly described. This is not acceptable for a systems biology paper. The proteomic analysis should include a multiple comparison correction like Benjamini Hochberg (Fig S3/S4). For S2F the "n" should be 3 mice and not 90 cells. These improper methods unfairly inflate the level of significance. Was normal distribution of the data tested? Several figures lack information on the number of replicates whether SD or SEM is shown (e.g. Fig 4, 6). According to Nature policy box plots should be shown instead of bar graphs throughout the manuscript.

-Fig 1: Finding substantial levels of TDP-43 pathology in a heterozygous PGRN mouse would be quite impressive. This has to be confirmed by fractionation and western blot analysis of pTDP-43, total TDP-43 and c-terminal fragments, because right now we see only single aggregates. Double staining with DAPI and total TDP-43 should be performed to analyze nuclear clearance. How was Fig 1K quantified? What does + and ++ mean? It would be better to show the number of inclusions per field of view. Is only TDP-43 or pTDP-43 pathology shown? How do the ubiquitin-positive TDP-43 negative inclusions in the hippocampus at 12 months look like? Critical, because this region is probably involved in the memory phenotype.

-S3B/C: what can these graphs tell us? Is this an interaction network of differentially phosphorylated proteins? VCP/TDP/CHMB2B should be shown individually or removed.

-Fig S4: All quantified phospho-peptides should be included in the manuscript including proper statistics. Was increased TDP-43 phosphorylation detected by MS?

-Fig S5: These effects should be quantified.

-Page 10: the data on MARCKS phosphorylation should be shown

-Fig 3D-J: The Biacore method is not described in the manuscript. How are the receptor and ligands expressed. Is full length PGRN used or cleaved granulins? Confirmed interaction using an independent method, e.g. co-immunoprecipitation.

-Fig 4: The effects on Tyro3 phosphorylation in KI mice are nice. Were the relevant peptides also identified and quantified in the MS screen?

-Fig 4D-F: The graphs should be fully labeled. The GAPDH, Tyro3 and many other "total"-blots are overexposed which makes quantification results questionable. Lower exposures should be provided and used for quantification or these crucial experiments have to be repeated. Similar poor quality blots are shown in many panels of Fig 6.

-The PGRN/Gas6/Tyro3/Tau axis needs to be strengthened. Is Tyro3 knockdown blocking the effects on GRN/Gas6 on Shc, b-raf, PLCgamma and PKCalpha? Does Gas6/PGRN affect Tau phosphorylation? Can the presumed effect of Gas6/GRN on Tau phosphorylation be prevented by Tyro3 knockdown?

-Fig 6/S6/S7: Vemurafenib inhibits b-raf V600E at least 50x more potently than wildtype in cellular assays and in fact may even activate wild-type b-raf (Holderfield et al, Nat Rev Cancer 2014).

Therefore it is unclear whether it is effective on wildtype b-raf in the experiments in Fig 6, S6 and S7. The authors have to show that it inhibits b-raf at their dosage in wildtype mice. While Vemurafenib may have a small effects on wildtype b-raf in the KI mice at high concentration, a high dosage might increase off-target effects on many other kinases (e.g. ZAK/JNK, Vin et al, Elife 2014). Full b-raf inhibition would probably cause a dramatic decrease in pMEK1, which is not seen in Fig 6C. That's probably the reason why the drug is licensed only for patients with the b-raf V600E mutation. The specificity of the treatment has to be addressed.

-Fig 6/S7: The authors show that Vermurafenib and Go6976 have no immediate effect on TDP-43 aggregation, but might they inhibit TDP-43 phosphorylation at Ser409/410?

-Fig 7: The error bars are surprisingly tight for an in vivo experiment. What is "N" in these experiments? N should be the number of mice analyzed, not the number of spines, dendrites or images. How many spines or what length of dendrite was analyzed for each mouse? The unit for spine number is most certainly incorrect, because 8-10 spines cannot fit into 1 um of dendrite.

Reviewer #2 (Remarks to the Author):

Summary

- Authors generated and extensively characterized a new PGRN mouse KI model to study FTLTDP (or FTLTDP-U) (frontotemporal lobar dementia type TDP (ubiquitin and TDP-43 positive, tau negative dementia); this mouse model is very useful to the field as previously generated KO models of PGRN failed to recapitulate the human pathology of the disease. The authors should be applauded for their use of many different and complementary technologies and approaches and many of these results are interesting and novel.

- Interesting story and research question with some very high quality data and new and potentially important conclusions.

- Unfortunately the manuscript is a somewhat poorly written and sub-optimally assembled, I found it very hard to read and follow; if considered for publication, the text must be significantly improved and the order / number of figures as well as content must be reconsidered (tightened). Overall I like this manuscript and believe it's findings are important and will be of interest to the field.

Comments

- I think this manuscript deserves to be published at nature communications if the authors can reduce the amount of data presented, significantly improve the text, and tighten the message.

1. Overall there is just too much data and the authors discuss supplementary figures before discussing main figures. The manuscript should be revised so that 1) figures are addressed in order 2) supplementary data contain only data that support main figures. I would move most of figure one to the sup and start with figure two.

2. The mass spectrometry is well done and the findings are well supported.

3. Some of the western blots are excellent and others not so. For example, how did the authors perform the analysis of WB given that majority of GAPDH control bands are not separated on the representative blots? Also, the loading is uneven in some of the panels

4. In theory it would be nice if Fig 6: panels A-E, H-L – all of the panels had an additional control B6+Vemurafenib. Without this control the potential specificity of this finding is a bit difficult to assess.

5. For the bar graphs in Figures 1H and 6F. Are these measurement made from the same set of data? I am not sure if the matters much.

6. Figure 7: ideally rescue experiments would also include drug treatment of WT mice.

7. 7C – PBS treatment of PGRN-KI mouse caused dramatic morphological changes to the spines this raises concerns regarding the confidence of these findings if even the negative controls are fluctuating this much. I would suggest just swapping the image panels so that the 2 representative images are more similar. I believe the quantitation.

8. The mass spec experiments need to be described a bit further in the main text. Was a post-hoc correction used in the calculation of the p values?

9. Control experiments with phosphatase spiked into the extract should be included, this way the

specificity of the phospho-bands could be confirmed (Fig 4 D-F).

10. Figure 5 is already convincing and mutation of the sites to alanine for additional control could maybe be added.

11. Figure 8 is excellent.

12. Manuscript does not conform to all of the Nature manuscript guidelines

Minor comments

- The title mentions synapses, but I only see post-synaptic staining in the IHC panels?
- Figure 1A: GAPDH loading controls do not match up to the PGRN blots (different widths).
- Figure 1A: R&B and Calbiochem antibodies against PGRN detect bands of different sizes, is this expected?
- Figure 3B: It is unclear which experiments were performed on which brain region i.e. does "cerebral cortex" always refer to frontal cortex in general? Some figures list M2, while others specify RSD. Authors should clarify and at least keep nomenclature consistent so the readers can easily follow.
- Authors are not consistent with figure labeling: some experiments have the N listed, some do not, similarly: p-values, WB ladders, blot framing, font sizes etc. some blots are tiny.
- Figure 3C, could more healthy cells be shown?
- Figure 4 E - unequal loading of sample?
- Figure 1f can be removed or show control with zoom.
- Figure 6: panel H is never mentioned in the text
- PSD staining in Figure 7H seems a bit odd, are the authors sure of this staining pattern?
- Figure S3B-C needs more explanation or labels, I get the point but many reads will not.

Jeffrey N. Savas

Reviewer #3 (Remarks to the Author):

The manuscript by Fujita et al. described intriguing phenotypes of a PGRN knockin model. They found that PGEN could inhibit Gas6-dependent activation of TYRO3. They performed phosphoproteome analysis, and provided evidence of disinhibition of Gas6 binding to TYRO3 by decreased PGRN activates PKC α through PLC γ , which is associated with tau phosphorylation at Ser203. Using a PKC inhibitor, b-raf inhibitor or knockdown of tau, they showed amelioration of spine loss and cognitive impairment of PGRN-KI mice. There are a few concerns that need to be addressed:

- 1) The inhibition of PGRN on Gas6-Tyro3 pathway is very interesting. In order to conclude that disinhibition of Gas6-Tyro3 underlies the downstream signaling and toxicity, it would be important to show that Tyro3 or Gas6 reduction ameliorates the phenotype PGRNKI neurons, on tau phosphorylation etc.
- 2) B-Raf and PKC have many substrates. It is not possible to directly link the protective effects of B-Raf and PKC inhibitors on Morris water maze with tau phosphorylation. A more direct evidence would be cross PGRNKI with tauKO, with or without the inhibitors. The protective effects of shTau on spine loss does not support their conclusion that "the rescuing effects of PKC and b-raf inhibition on cognitive function of PGRN-KI mice were mediated by mislocalization of pSer203-tau.
- 3) The R504Stop results in reduction of PGRN protein and mRNA by 50%. It is important to perform direct comparison of some of the key observations. Regardless of the outcome, it is critical to assess whether the alternations depend on PGRN levels per se or other processes, such as activation of non-sense mediated decay.

Minor

- 1) What age were the spine measurements done?
- 2) No methods on the binding assay between PGRN and Gas6
- 3) Controls to show the p-tau stainings are specific?

Reviewers' comments:

Reviewer #1 (Remarks to the Author):

The manuscript by Fujita et al uses a variety of methods to uncover a potential new pathway in PGRN-mediated FTLD. First, they developed a knockin mouse model for progranulin, which surprisingly shows TDP-43 pathology. Second, they use phospho-proteomics to analyze altered signaling in these mice. Third, this leads to discovery of TYRO3 a putative novel PGRN-receptor, or more precisely a ligand/receptor pair inhibited by PGRN. Fourth, they use two candidate drugs to prevent altered signaling in the mouse model, which apparently improves cognition. One of the drugs, Vemurafenib, is FDA approved for cancer with the V600E b-raf mutation, because it is thought to be specifically acting on the mutant protein. There it is unclear if the drug is safe at a dose necessary for efficacy in mice/patients with wild-type b-raf. Cramming this much data in one paper is impressive, but also prevents a thorough analysis of many crucial points and thus hurts the manuscript overall.

>>> We appreciate very much the reviewer for kind evaluation of our paper.

>>> Regarding the concern about safety of the drug, our protocol is oral administration of 32mg/body weight kg/day for 6 weeks every day, which corresponds to the recommended dose for human melanoma patients (https://www.accessdata.fda.gov/drugsatfda_docs/label/2011/202429s000lbl.pdf). In rat and dog oral administration of 1000mg/body weight kg everyday for 1 month was tested, and there was no side effect according to the Drug package insert regarding Vemurafenib distributed from the pharmaceutical company in Japan.

We do NOT intend to introduce Vemurafenib directly into clinical application. But considering with the effective dose in this study, we think it might be possible to use certain safer and more effective derivatives (if available in the near future) for clinical trials of human patients.

-The manuscript is readable but would greatly benefit from English language editing. There are many typos (e.g. "PGN" and "moths") and weird sentences. Some words are used in the wrong context and far too frequently (e.g. "obviously", "actually").

>>> We asked a professional editor to improve our manuscript. We also corrected typos.

-Statistical analysis is weak and poorly described. This is not acceptable for a systems biology paper. The proteomic analysis should include a multiple comparison correction like Benjamini Hochberg (Fig S3/S4).

>>> We replaced old data in S3 and S4 with new results evaluated by Welch's test with post-hoc BH procedure. All the phosphoproteome data were re-examined by using q-value, and we reconfirmed our conclusion that b-raf signaling pathway was activated (new Figure S4 and database at <http://suppl.atgc.info/011/>). Throughout figure legends, descriptions about statistics were improved.

For S2F the "n" should be 3 mice and not 90 cells. These improper methods unfairly inflate the level of significance. Was normal distribution of the data tested? Several figures lack information on the number of replicates whether SD or SEM is shown (e.g. Fig 4, 6).

>>> We corrected the silly number of samples for the graph, and it is now N=3 using the mean values of 30 cells groups. We confirmed the normal distribution of signal intensities obtained from 90 cells. We also checked the normal distribution in the other data when necessary. We added the description "Average and S.E.M. are shown" to figure legends when necessary.

According to Nature policy box plots should be shown instead of bar graphs throughout the manuscript.

>>> We asked the editor about the way of response to this reviewer's comment for presentation of graphs. Following the advice of the editor, we preserved the graph style and attached the original data for the graphs in Supplementary Information.

-Fig 1: Finding substantial levels of TDP-43 pathology in a heterozygous PGRN mouse would be quite impressive. This has to be confirmed by fractionation and western blot analysis of pTDP-43, total TDP-43 and c-terminal fragments, because right now we see only single aggregates.

>>> We added western blots of total TDP-43 and pTDP-43 (pS409/410) in Figure 1L.

Double staining with DAPI and total TDP-43 should be performed to analyze nuclear clearance.

>>> We added IHC to show nuclear reduction and cytoplasmic translocation of TDP43 (Figure 1I).

How was Fig 1K quantified? What does + and ++ mean? It would be better to show the number of inclusions per field of view.

>>> "+" means 1 ~ 4 positive cells / field (430 x 550 microM). "++" means 5 or more / field (430 x 550 microM). We added the description in figure legend.

We had considered showing the number of inclusion with mean +/- SEM in the panel following the advice of the reveiwer, but it was to busy. Thus we kept the panel in the current style and added explanation of the meaning of "+" and "++".

Is only TDP-43 or pTDP-43 pathology shown? How do the ubiquitin-positive TDP-43 negative inclusions in the hippocampus at 12 months look like? Critical, because this region is probably involved in the memory phenotype.

>>> We made mistake regarding the hippocampus at 12 months in generation of the panel. We corrected it to be positive in TDP43.

-S3B/C: what can these graphs tell us? Is this an interaction network of differentially phosphorylated proteins? VCP/TDP/CHMB2B should be shown individually or removed.

>>> S3B was now removed. S3C is a critical molecular network based on the molecules whose phosphorylation was changed at more than one site within the proteins. We kept the S3C (which was now re-numbered as S2B) and added explanation. Following the comment about the multiple group comparison of the data, we now re-generated the networks again based on q-value in Welch's test with post-hoc BH procedure, instead of previous p-value.

-Fig S4: All quantified phospho-peptides should be included in the manuscript including proper statistics. Was increased TDP-43 phosphorylation detected by MS?

>>> We added the data of the quantified phospho-peptides results with Welch's test with BH procedure to the manuscript text and figures as much as possible. Since the total data was too large to submit, we deposited all the original data to <http://suppl.atgc.info/011/> (ID: npat011, Password: hX1EKtk8).

>>> Mass spec analysis detected phosphorylated TDP43 as shown in new Figure S3C. But the detection rate was very low, possibly due to the aggregating characteristics of TDP43. The common pathological phosphorylation sites (pSer409/410) were not detected by phospho-protein mass spec unfortunately.

-Fig S5: These effects should be quantified.

>>> We added the results of quantitative analyses of signal intensities in IHC.

-Page 10: the data on MARCKS phosphorylation should be shown

>>> We added the data of phospho-MARCKS to Figure S3C.

-Fig 3D-J: The Biacore method is not described in the manuscript. How are the receptor and ligands expressed. Is full length PGRN used or cleaved granulins?

>>> We added the details of the Biacore method (Figure 3D-J) and the cell-based binding assay in Figure 3C to "Method section" of the manuscript. Full-length PGRN was used.

Confirmed interaction using an independent method, e.g. co-immunoprecipitation.

>>> We performed pull-down assay and added the results (Fig 2K).

-Fig 4: The effects on Tyro3 phosphorylation in KI mice are nice. Were the relevant peptides also identified and quantified in the MS screen?

>>> Tyro3 phosphorylation was detected by MS analysis, and we added the result to new Figure S4A.

-Fig 4D-F: The graphs should be fully labeled.

>>> We moved the graph to the position under the WB panels so that readers can understand the correspondence.

The GAPDH, Tyro3 and many other "total"-blots are overexposed which makes quantification results questionable. Lower exposures should be provided and used for quantification or these crucial experiments have to be repeated.

>>> We showed lower exposure of the panels and in some cases replaced the blot images with new ones, since the lower exposure data were not available. The graphs were also corrected in some cases with new data added during revision.

Similar poor quality blots are shown in many panels of Fig 6.

>>> We added new data to Fig 6 following the comment of reviewer 2. Thus we replaced with new data with better qualities.

-The PGRN/Gas6/Tyro3/Tau axis needs to be strengthened. Is Tyro3 knockdown blocking the effects on GRN/Gas6 on Shc, b-raf, PLCgamma and PKCalpha? Does Gas6/PGRN affect Tau phosphorylation? Can the presumed effect of Gas6/GRN on Tau phosphorylation be prevented by Tyro3 knockdown?

>>> We added results of requested experiments to Figure 4G. All the expectations of the reviewer were confirmed.

-Fig 6/S6/S7: Vemurafenib inhibits b-raf V600E at least 50x more potently than wildtype in cellular assays and in fact may even activate wild-type b-raf (Holderfield et al, Nat Rev Cancer 2014). Therefore it is unclear whether it is

effective on wildtype b-raf in the experiments in Fig 6, S6 and S7. The authors have to show that it inhibits b-raf at their dosage in wildtype mice. While Vemurafenib may have a small effects on wildtype b-raf in the KI mice at high concentration, a high dosage might increase off-target effects on many other kinases (e.g. ZAK/JNK, Vin et al, Elife 2014). Full b-raf inhibition would probably cause a dramatic decrease in pMEK1, which is not seen in Fig 6C. That's probably the reason why the drug is licensed only for patients with the b-raf V600E mutation. The specificity of the treatment has to be addressed.

>>> Thank you very much for your kind advice and critical information in previous literatures. The figure below is from the paper suggested by the reviewer (Holderfield et al, Nat Rev Cancer 2014). Fig. A is the situation of b-raf V600E mutant and Fig. B is the situation of normal b-raf, as you know well.

In mutants, b-raf is active at the default state, and an inhibitor can suppress mutant b-raf activity in a dose dependent manner. Meanwhile, normal b-raf molecules are inhibiting each other at the default state, and addition of an inhibitor releases the self-suppression of b-raf, thus paradoxically activates the activity of b-raf. A higher concentration of the inhibitor suppresses the b-raf activity.

It is intriguing, however, that the concentration for paradoxical activation and the concentration for suppression are different in each type of inhibitor. The following figure is borrowed from the paper by Bollag et al (*Nature* **467**, 596–599 (30 September 2010) doi:10.1038/nature09454), indicating that 100 nM of Vemurafenib (PLX4032) inhibits kinase activity of WILD-TYPE B-RAF, which is only three folds of the concentration inhibiting b-raf-V600E. We added the description in the result section of the text to prevent misunderstanding that the concentration used in this study was very high. It was actually an ordinary dose for melanoma patients.

Supplementary Table 1. Biochemical IC₅₀ determinations of the kinase inhibitory activity of PLX4032 versus a panel of kinases

Assay	IC ₅₀ nM*
B-RAF-V600E	31
C-RAF	48
B-RAF	100
SRMS	18
ACK1	19
MAP4K5 (KHS1)	51
FGR	63
LCK	183
BRK	213
NEK11	317
BLK	547
LYNB	599
YES1	604
WNK3	877
MNK2	1717
FRK (PTK5)	1884
CSK	2339
SRC	2389

>>> On the other hand, as the reviewer pointed out, the specificity or off-target effect remains as an issue. This is a problem related to all inhibitors. Actually, in

the same list of Bollag et al shown above, a lower concentration (< 100nM) of vemurafenib could suppress ACK1, MAP4K5, FGR in addition to ZAK/JNK that the reviewer mentioned.

Therefore, we actually performed phosphoproteome analysis of Vemurafenib-treated KI mice v.s. non-treated KI mice (Figure S5), and analyzed the range and impact of the off-target effect that specifically occurs in our protocol (Figure S5B). As expected by the reviewer (also by us), Vemurafenib is of course not perfectly selective, but relatively selective to b-raf pathway according to the data presented in Figure S5A&S5B. Also the effect on b-raf on downstream molecules was sufficiently wide-ranged as shown in Figure S5A. These results suggest that even at our administration dose Vemurafenib could suppress b-raf and downstream molecules to an extent, which is sufficient for neurodegeneration phenotype though might be insufficient for melanoma without V600E mutation.

Other pathways-mediated rescue might a concern. However, the weakness in our inhibitor experiments is now complemented with b-raf KD experiments with behavior tests and spine morphology (Fig 6) in this version.

-Fig 6/S7: The authors show that Vermurafenib and Go6976 have no immediate effect on TDP-43 aggregation, but might they inhibit TPD-43 phosphorylation at Ser409/410?

>>> MS analysis of Vemurafenib-treated KI mice did not detect sufficient amount of TDP43 proteins (including phospho-TDPs) as shown in Figure S3C. Therefore we performed immunohistochemistry of PGRN-KI mice with anti pSer409/410 antibody, and confirmed that the numbers of pSer409/410-positive cells as well as of TDP43 aggregation-positive cells were not changed (Figure S6).

-Fig 7: The error bars are surprisingly tight for an in vivo experiment. What is "N" in these experiments? N should be the number of mice analyzed, not the number of spines, dendrites or images. How many spines or what length of dendrite was analyzed for each mouse?

>>> Previous analysis was performed defining number of dendrite shafts observed by two-photon microscopy as N. Actual number of mice was three, so we generated the mean value of a single mouse and used three values for statistical analyses.

The unit for spine number is most certainly incorrect, because 8-10 spines cannot fit into 1 um of dendrite.

>>> We thank the reviewer's comment. We checked the analysis and confirmed the calculation was wrong. We corrected the presentation of the unit.

Reviewer #2 (Remarks to the Author):

Summary

- Authors generated and extensively characterized a new PGRN mouse KI model to study FTLN-TDP (or FTLN-U) (frontotemporal lobar dementia type TDP (ubiquitin and TDP-43 positive, tau negative dementia); this mouse model is very useful to the field as previously generated KO models of PGRN failed to recapitulate the human pathology of the disease. The authors should be applauded for their use of many different and complementary technologies and approaches and many of these results are interesting and novel.

- Interesting story and research question with some very high quality data and new and potentially important conclusions.

>>> Thank you so much for the kind evaluation.

- Unfortunately the manuscript is a somewhat poorly written and sub-optimally assembled, I found it very hard to read and follow; if considered for publication, the text must be significantly improved and the order / number of figures as well as content must be reconsidered (tightened). Overall I like this manuscript and believe it's findings are important and will be of interest to the field.

>>> We completely accept the criticism and advice from the reviewer. We did all efforts to tighten the manuscript and improved the order. First, we delete the figures about inflammation and DNA damage (previous Fig S2A-F and previous Fig S5B, C), which will be probably included in another paper in the future. Correspondingly we deleted a section in Introduction related to inflammation and TNF-PGRN relationship. Second, we deleted negative results of behavioral tests from Figures (previous Fig S1). Third, there were many redundant sections and descriptions in previous version, which we deleted them from the text. By the reconstruction, even though new data were added to revised version, we reduced one Figure, one Sup Figure, and reduced the pages of text.

Comments

- I think this manuscript deserves to be published at nature communications if the authors can reduce the amount of data presented, significantly improve the text, and tighten the message.

>>> As described above, we deleted an introduction paragraph and data related to inflammation from this manuscript. The removal made the message of this paper to be simple, we believe. We also deleted some unessential figure such as Figure S3B following the request of reviewer #1.

1. Overall there is just too much data and the authors discuss supplementary figures before discussing main figures. The manuscript should be revised so that
1) figures are addressed in order 2) supplementary data contain only data that

support main figures. I would move most of figure one to the sup and start with figure two.

>>> Following the advice of the reviewer, we moved behavioral analyses of old Figure 1 to Supplementary Figure 1, and combined the pathological analyses of TDP43/Ub with old Figure 2, and made new Figure 1. We also deleted negative data in old Sup Figure 1 and just mentioned in the text. With these changes, we now start with the main Figure.

2. The mass spectrometry is well done and the findings are well supported.

>>> Thank you so much.

3. Some of the western blots are excellent and others not so. For example, how did the authors perform the analysis of WB given that majority of GAPDH control bands are not separated on the representative blots? Also, the loading is uneven in some of the panels

>>> Thank you for the important comment. Actually the bands of GAPDH were fused and the margin on each lane was hard to define. In the previous version we used the notches in such fused bands as the marker to separate the areas for quantification. But responding to the comment, we repeated the blots and replaced with better ones showing separated GAPDH bands. We corrected the panels whose loadings seemed uneven, and quantification was repeated with the new data.

4. In theory it would be nice if Fig 6: panels A-E, H-L – all of the panels had an additional control B6+Verumrafenib. Without this control the potential specificity of this finding is a bit difficult to assess.

>>> We followed the advice and added B6+Vemurafenib as the control.

5. For the bar graphs in Figures 1H and 6F. Are these measurement made from the same set of data? I am not sure if the matters much.

>>> Thank you for the comment. We checked the situation and confirmed that they were the same data. We agree it is better to repeat the same set of experiment independently for different figures, so we now performed new (but the same) experiments and made the new figures.

6. Figure 7: ideally rescue experiments would also include drug treatment of WT mice.

>>> We added the results of requested experiments to Figure 7.

7. 7C – PBS treatment of PGRN-KI mouse caused dramatic morphological changes to the spines this raises concerns regarding the confidence of these findings if even the negative controls are fluctuating this much. I would suggest just swapping the image panels so that the 2 representative images are more similar. I believe the quantitation.

>>> We appreciate very much the kind advice. We changed the panels so that representative images look consistent.

8. The mass spec experiments need to be described a bit further in the main text. Was a post-hoc correction used in the calculation of the p values?

>>> We added brief description about the method of proteome analyses in the main text. In the new version we employed q-value in Welch's test with post-hoc BH procedure, and renewed all the related data throughout the manuscript.

9. Control experiments with phosphatase spiked into the extract should be included, this way the specificity of the phospho-bands could be confirmed (Fig 4 D-F).

>>> We performed the requested experiments with phosphatase and added the data to new Figure 3D-G. Thanks to the results we confirmed the specificity of the phospho-protein bands.

10. Figure 5 is already convincing and mutation of the sites to alanine for additional control could maybe be added.

>>> We added GST/EGFP-tau with alanine mutations (S214A and T231A) as controls for in vitro and in vivo kination experiments.

11. Figure 8 is excellent.

>>> Thank you very much.

12. Manuscript does not conform to all of the Nature manuscript guidelines

>>> We followed the manuscript guidelines and corrected all the points we noticed.

Minor comments

- The title mentions synapses, but I only see post-synaptic staining in the IHC panels?

>>> We appreciate the criticism, but the post-synapse is the critical component of synapse. So if it is not a big issue, we would appreciate if the reviewer kindly allows us to use the title.

- Figure 1A: GAPDH loading controls do not match up to the PGRN blots (different widths).

>>> We replaced the panels with new ones (Figure S1A).

- Figure 1A: R&B and Calbiochem antibodies against PRGN detect bands of different sizes, is this expected?

>>> This is an old experiment that one of the co-authors did. We repeated again for this revision and changed a batch of the antibody. We found the results were basically similar between two antibodies, and we think the new data are more reasonable.

- Figure 3B (Author: does it mean Fig 2B?) : It is unclear which experiments were performed on which brain region i.e. does “cerebral cortex” always refer to frontal cortex in general? Some figures list M2, while others specify RSD. Authors should clarify and at least keep nomenclature consistent so the readers can easily follow.

>>> We apologize for the confusing descriptions about brain regions examined in this study. We mean “cerebral cortex” as whole cerebral cortex including frontal, parietal, occipital and temporal lobes. In previous Fig 2B (new Fig1F), we used whole cerebral cortex because fresh cerebral samples are hard to separate specifically to each lobe.

>>> We described more precisely which region was injected and which region was observed by two-photon microscopy. Basically all experiments are focusing on frontal lobe, especially on M2 region. However, injection should be adjacent but not the exactly the same position for observation, because injection damages brain tissue. Basically we inject vectors into RSD and observed spines in M2. We described it more precisely in related parts.

>>> We consistently kept nomenclatures, such as M2, FrA and RSD in this new version.

- Authors are not consistent with figure labeling: some experiments have the N listed, some do not, similarly: p-values, WB ladders, blot framing, font sizes etc. some blots are tiny.

>>> We improved the figure labeling throughout the manuscript following the advices of the reviewer. P-values and the statistical methods are deleted from Figures, and original data for all graphs, statistical tests, N and p-values are shown in the list of Supplementary Information. N and statistics methods are also described in Figure legends. In exceptional cases when numbers (N) were not equalized among the groups for graph, we described the numbers in graph bars. We used asterisks for student's t-test, while we used sharp marks (#, ##) for Dunnet's test and Tukey's HSD test. We tried to make blots to be sufficient sizes in figures.

- Figure 3C, could more healthy cells be shown?

>>> We replaced the panels with new ones

- Figure 4 E – unequal loading of sample?

>>> We replaced the panels with representative ones.

- Figure 1f can be removed or show control with zoom.

>>> We added control with zoom (Figure S1G).

- Figure 6: panel H is never mentioned in the text

>>> We corrected the error.

- PSD staining in Figure 7H seems a bit odd, are the authors sure of this staining pattern?

>>> We also felt the size of PSD stains were little bit large. However, we repeated the experiments and we also used the same antibody for other purposes (for other projects). From the experiences, we are sure about it.

- Figure S3B-C needs more explanation or labels, I get the point but many reads will not.

>>> We appreciate the kind advice. We added more detailed explanation about the panels in figure legends and also in the text. Meanwhile, Figure S3B was deleted following the request of reviewer #1.

Jeffrey N. Savas

Reviewer #3 (Remarks to the Author):

The manuscript by Fujita et al. described intriguing phenotypes of a PGRN knockin model. They found that PGEN could inhibit Gas6-dependent activation of TYRO3. They performed phosphoproteome analysis, and provided evidence of disinhibition of Gas6 binding to TYRO3 by decreased PGRN activates PKC α through PLC γ , which is associated with tau phosphorylation at Ser203. Using a PKC inhibitor, b-raf inhibitor or knockdown of tau, they showed amelioration of spine loss and cognitive impairment of PGRN-KI mice. There are a few concerns that need to be addressed:

1) The inhibition of PGRN on Gas6-Tyro3 pathway is very interesting. In order to conclude that disinhibition of Gas6-Tyro3 underlies the downstream signaling and toxicity, it would be important to show that Tyro3 or Gas6 reduction ameliorates the phenotype PGRNKI neurons, on tau phosphorylation etc.

>>> We added knockdown experiments of Tyro3 and Gas 6 in Figure 6. We examined spine morphology, tau phosphorylation, abnormal colocalization of p-Tau and PSD95, reduction of PSD95 spots, and memory/cognition tests.

2) B-Raf and PKC have many substrates. It is not possible to directly link the protective effects of B-Raf and PKC inhibitors on Morris water maze with tau phosphorylation. A more direct evidence would be cross PGRNKI with tauKO, with or without the inhibitors. The protective effects of shTau on spine loss does not support their conclusion that “the rescuing effects of PKC and b-raf inhibition on cognitive function of PGRN-KI mice were mediated by mislocalization of pSer203-tau.

>>> We agree with the advice that crossing with tauKO mice would be a better proof. But it takes a very long time to complete the experiments. Therefore we consulted with the editor and received the following answer.

“--- Regarding the double tau-KO PGRN-KI request I appreciate your concern, and I think that using shRNA for knockdown of tau and assessing performance in water maze using the b-raf inhibitor PKC inhibitor could be a useful alternative to address this issue. ---“

We followed the advice of the editor, and added the water-maze test and elevated plus maze test of tau-KD mice, with or without b-raf inhibitor (Vemurafenib) and PKC inhibitor (Go6976). The results are shown in Figure 6O.

3) The R504Stop results in reduction of PGRN protein and mRNA by 50%. It is important to perform direct comparison of some of the key observations. Regardless of the outcome, it is critical to assess whether the alternations depend on PGRN levels per se or other processes, such as activation of non-sense mediated decay.

>>> We checked the amount of PGRN mRNA in PGRN-KI mouse-derived primary neurons after blocking non-sense mediated RNA decay by UPF1-siRNA or CHX, and found it was normalized basically (Figure S1C). This means that the reduction is mostly due to RNA decay rather than decreased transcription.

Minor

1) What age were the spine measurements done?

>>> We apologize for unclearness. It was 12 weeks of age and we showed protocols in Figure 6A and the legend.

2) No methods on the binding assay between PGRN and Gas6

>>> We added a section in "Methods" for the binding assay between PGRN and Gas6

3) Controls to show the p-tau stainings are specific?

>>> We performed antibody absorption with the antigen (each phosphopeptides) and showed the results in Fig 1E. We also kept C57BL/6J as a negative control for such anti-phospho-Tau antibodies. The results support the p-tau stainings are specific.

REVIEWERS' COMMENTS:

Reviewer #1 (Remarks to the Author):

The manuscript greatly improved in terms of science and readability and is now suited for publication in NCOMMS.

Reviewer #2 (Remarks to the Author):

The authors have done an impressive job responding to my comments and the revised manuscript looks fantastic. The figures look great and the text is very nice. Overall the conclusions are very well supported by the data and I believe it should now be accepted at NC.

Jeffrey Savas

Reviewer #3 (Remarks to the Author):

The revised manuscript is much improved. The authors have addressed this reviewer's comments adequately.

REVIEWERS' COMMENTS:

Reviewer #1 (Remarks to the Author):

The manuscript greatly improved in terms of science and readability and is now suited for publication in NCOMMS.

Reviewer #2 (Remarks to the Author):

The authors have done an impressive job responding to my comments and the revised manuscript looks fantastic. The figures look great and the text is very nice. Overall the conclusions are very well supported by the data and I believe it should now be accepted at NC.

Jeffrey Savas

Reviewer #3 (Remarks to the Author):

The revised manuscript is much improved. The authors have addressed this reviewer's comments adequately.

>>>>We thank all reviewers for their critical evaluation and kind efforts.